# ITERATIVE RELAXING GRADIENT PROJECTION FOR CONTINUAL LEARNING

## ABSTRACT

A critical capability for intelligent systems is to continually learn given a sequence of tasks. An ideal continual learner should be able to avoid catastrophic forgetting and effectively leverage past learned experiences to master new knowledge. Among different continual learning algorithms, gradient projection approaches impose hard constraints on the optimization space for new tasks to minimize task interference, yet hinder forward knowledge transfer at the same time. Recent methods use expansion-based techniques to relax the constraints, but a growing network can be computationally expensive. Therefore, it remains a challenge whether we can improve forward knowledge transfer for gradient projection approaches *using a fixed network architecture*. In this work, we propose the Iterative Relaxing Gradient Projection (IRGP) framework. The basic idea is to iteratively search for the parameter subspaces most related to the current task and relax these parameters, then reuse the frozen spaces to facilitate forward knowledge transfer while consolidating previous knowledge. Our framework requires neither memory buffers nor extra parameters. Extensive experiments have demonstrated the superiority of our framework over several strong baselines. We also provide theoretical guarantees for our iterative relaxing strategies.

## 1 INTRODUCTION

A critical capability for intelligence systems is to continually learn given a sequence of tasks (Thrun & Mitchell, 1995; McCloskey & Cohen, 1989). Unlike human beings, vanilla neural networks straightforwardly update parameters regarding current data distribution when learning new tasks, suffering from catastrophic forgetting (McCloskey & Cohen, 1989; Ratcliff, 1990; Kirkpatrick et al., 2017). As a result, continual learning is gaining increasing attention in recent years (Kurle et al., 2019; Ehret et al., 2020; Ramesh & Chaudhari, 2021; Liu & Liu, 2022; Teng et al., 2022). An ideal continual learner is expected to not only avoid catastrophic forgetting but also facilitating forward knowledge transfer (Lopez-Paz & Ranzato, 2017), which is to leverage past learning experiences to master new knowledge efficiently and effectively (Parisi et al., 2019; Finn et al., 2019).

Several types of methods have been proposed for continual learning. Replay-based methods (Lopez-Paz & Ranzato, 2017; Shin et al., 2017) alleviate catastrophic forgetting by storing some old samples in the memory as they are inaccessible when new tasks come, while expansion-based methods (Rusu et al., 2016; Yoon et al., 2017; 2019) expand the model structure to accommodate incoming knowledge. However, these methods require either extra memory buffers (Parisi et al., 2019) or a growing network architecture as new tasks continually arrive (Kong et al., 2022), which always results in expensive computation costs (De Lange et al., 2021). In order to maintain a fixed network capacity, regularization-based methods (Kirkpatrick et al., 2017; Zenke et al., 2017; Aljundi et al., 2018) penalize the transformation of parameters regarding the corresponding plasticity via regularization terms. While these regularization terms are applied to individual neurons, recent gradient projection methods (Zeng et al., 2019; Saha et al., 2021; Wang et al., 2021) modify the gradients in the feature space by constraining the directions of gradient update, which achieves outstanding performance.

However, although gradient projection methods effectively mitigate forgetting within a fixed network capacity (Zeng et al., 2019), the capability of learning new tasks is hindered by the limited optimization space, resulting in insufficient forward knowledge transfer. In other words, constraining the directions of gradient update fails on the plasticity in the stability-plasticity dilemma (French, 1997).

Figure 1: Illustration of our proposed IRGP method and two baselines: GPM and TRGP. Blocks painted in different colors denote the parameters optimized after different tasks. We denote the relaxing subspace within the frozen space as the painted stripes in our IRGP pipeline.

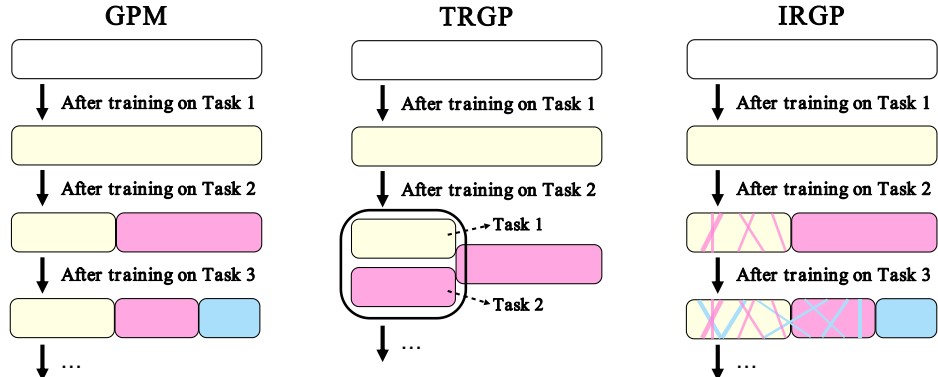

Trust Region Gradient Projection (Lin et al., 2022) tackles this problem by expanding the selected subspace of old tasks as trust regions with scaled weight projection, similar to other expansion-based methods (Yoon et al., 2019). In spite of substantial improvement, these methods are computationally expensive as a result of growing network architecture (Wang et al., 2021). Therefore, insufficient forward knowledge transfer remains a key challenge for gradient projection methods.

To address this challenge, we propose the Iterative Relaxing Gradient Projection (IRGP) framework to facilitate forward knowledge transfer within a fixed network capacity. We design a simple yet effective strategy to find the critical subspace within the frozen space. During the training phase, we iteratively reuse the parameters within the selected subspace. Instead of strictly freezing those parameters, our method explores a larger optimization space, which allows better forward knowledge transfer and thus achieves better performance on new tasks. The procedure of our approach is illustrated in Figure 1. Extensive experiments on various continual learning benchmarks demonstrate that our IRGP framework promotes forward knowledge transfer and achieves better classification performance compared with related state-of-the-art approaches. Moreover, our framework performs can also be extended as an expansion-based methods by storing the parameters of the selected relaxing subspace, universally surpassing TRGP (Lin et al., 2022) and other expansion-based approaches. We also provide theoretical proof to guarantee the efficiency of our relaxing strategy.

## 2 RELATED WORK

In this section, we review the representative approaches for continual learning and briefly analyze their differences from our method. Conceptually, these approaches can be roughly divided into the following four categories.

**Replay-based methods:** These methods maintain a complementary memory for old samples, which are replayed during learning novel tasks. GEM (Lopez-Paz & Ranzato, 2017) constrains gradients concerning previous samples and Chaudhry et al. (2018) further propose to estimate with random samples to accelerate. While past samples are commonly not accessible in the real world, auxiliary deep generative models are thus deployed to synthesize pseudo data (Chenshen et al., 2018; Cong et al., 2020). Recent approaches (PourKeshavarzi et al., 2021; Choi et al., 2021) leverage a single model for both classification and pseudo data generation. However, including extra data into the current task introduces excessive training time (De Lange et al., 2021), especially on long task sequence. Our approach requires no previous data, in other words, is a replay-free method.

**Expansion-based methods:** Expansion-based methods dynamically allocate new parameters or modules to learn new tasks. Rusu et al. (2016) propose to incrementally introduce additional subnetworks with a fixed capacity. DEN (Yoon et al., 2017) selectively retrains the frozen model and expands only with necessary neurons. Moreover, Li et al. (2019) perform an explicit network architecture search to decide where to expand. APD (Yoon et al., 2019) further decomposes the network and utilizes sparse task-specific parameters. However, these methods face capacity explosion inevitably after learning a long sequence of tasks. In contrast, our approach maintains a fixed network architecture to avoid expensive model growth.

**Regularization-based methods:** Methods in this category introduce extra regularization terms to the objective function to penalize the modification of parameters. EWC (Kirkpatrick et al., 2017) first proposes to constrain the change based on the importance weight approximated by Fisher Information Matrix. MAS (Aljundi et al., 2018) measures the importance of the sensitivity of model outputs under an unsupervised setting. Other methods, also called parameter-isolation methods, defy catastrophic forgetting via freezing the gradient updates of particular parameters (De Lange et al., 2021). PackNet (Mallya & Lazebnik, 2018) iteratively prunes and allocates parameters subset to corresponding tasks, whereas HAT (Serra et al., 2018) learns task-based hard attention to identify important parameters. Instead of restricting individual parameters with estimated importance, the main idea of our approach is constraining the direction of gradients.

**Gradient projection methods**: Gradient projection methods directly constrain the gradients to overcome catastrophic forgetting, and our approach belongs to this category. Mehta et al. (2021) implicitly expands the model with respect to the frozen space and GEM (Lopez-Paz & Ranzato, 2017) utilizes complementary memory to restrict the update. While our approach requires neither storing old samples nor expanding the network. OWM (Zeng et al., 2019) first proposed to modify the gradients upon projector matrices. OGD (Farajtabar et al., 2020) keeps the gradients orthogonal to the space spanned by previous gradients, whereas GPM (Saha et al., 2021) computes the frozen space based on old data. NCL (Kao et al., 2021) combines the idea of gradient projection and Bayesian weight regularization to mitigate catastrophic forgetting. In spite of minimizing backward interference, these approaches suffer poor forward knowledge transfer and lack plasticity (Kong et al., 2022). TRGP (Lin et al., 2022) expands the model with trust regions based on task relationship to achieve better performance on new tasks. In contrast, we focus on facilitating forward knowledge transfer within a fixed capacity network by iteratively relaxing frozen regions with constraints.

## 3 ITERATIVE RELAXING GRADIENT PROJECTION

### 3.1 PRELIMINARIES

In a continual learning setting, we consider $T$ tasks arriving as a sequence. The datasets are denoted as $\mathcal{D}^{(t)} = \{x_i^{(t)}, y_i^{(t)}\}_{i=1}^{N_t}$, where $N_t$ is the number of samples. When learning the current task, the datasets of old tasks are inaccessible. We use an $L$-layer neural network with fixed capacity, and parameters defined as $\mathcal{W} = \{W^l\}_{l=1}^{L}$, where $W^l$ denotes the parameters in the $l$-th layer. The model is optimized by minimizing the objective function (1) and $\mathcal{L}_t$ is the loss function for task $t$.

$$\mathcal{L}(\mathcal{W}, \mathcal{D}^{(t)}) = \frac{1}{N_t} \sum_{i=1}^{N_t} \mathcal{L}_t(f(x_i^{(t)}; \mathcal{W}), y_i^{(t)}) \tag{1}$$

Gradient projection methods mitigate catastrophic forgetting by only updating the model in the orthogonal direction to frozen spaces. Saha et al. (2021) proposed to compute the frozen spaces based on the inputs of each layer. For task $t$, the frozen gradient spaces for the first $t-1$ tasks are denoted as $\mathcal{U}_{t-1} = \{U_{t-1}^l\}_{l=1}^{L}$, where $U_t^l$ is the frozen space of layer $l$ for task $t$. During the training phase, for each layer $l$, gradients $g_t^l$ are constrained to be orthogonal to $U_{t-1}^l$. Particularly, assuming $\mathbf{B}_t^l = [u_{t-1,1}^l, ..., u_{t-1,N}^l]$ as the total $N$ basis for $U_{t-1}^l$, gradients $g_t$ are modified as:

$$g_t^l = g_t^l - \mathrm{Proj}_{U_{t-1}^l}(g_t^l) = g_t^l - g_t^l \mathbf{B}_{t-1}^l (\mathbf{B}_{t-1}^l)^T \tag{2}$$

After getting the learned model $\mathcal{W}_t = \{W_t^l\}_{l=1}^{L}$, for each layer $l$, we record the intermediate representation $h_{t,j}^l$ of the $j$-th input and stack them to obtain the representation matrix $\mathbf{H}_t^l = [h_{t,1}^l, ..., h_{t,N_t}^l]$. Then compress the representation matrices by performing Singular Value Decomposition $\mathrm{SVD}(\mathbf{H}_t^l) = \mathbf{U}_{H,t}^l \mathbf{\Sigma}_t^l (\mathbf{V}_t^l)^T$. Given the threshold $\epsilon_{th}^l$, select the first $k$ vectors in $\mathbf{U}_{H,t}^l$ based on the following criteria:

$$\|\mathbf{\Sigma}_t^l[0:k]\|_F^2 \geq \epsilon_{th}^l \|\mathbf{\Sigma}_t^l\|_F^2 \tag{3}$$

to construct the significant representation space $R_t^l = \mathrm{span}\{\mathbf{U}_{H,t}^l[0:k]\}$, where $\|\cdot\|_F^2$ denotes Frobenius norm here. The significant representation spaces, considered as the frozen spaces for current task $t$, are then merged into the whole frozen gradient spaces for the first $t$ tasks:

$$\mathcal{U}_t = \{U_t^l\}_{l=1}^{L} = \{U_{t-1}^l \cup R_t^l\}_{l=1}^{L} \tag{4}$$

Although freezing gradients update significantly mitigates catastrophic forgetting, limited optimization space hinders the forward knowledge transfer, compromising the performance of new tasks. TRGP (Lin et al., 2022) tackles this problem by selecting old tasks relevant to the current task and expanding the corresponding frozen spaces as the trust regions. The scaled weight projection is further designed for memory-efficient updating and storing the parameters within the trust regions by scaling the basis, instead of directly changing the parameters. Considering that task $i$ is selected as the trust region, the scaled weight projection is shown as:

$$\text{Proj}_{U_i^l}^{S_i^l}(g_t^l) = g_t^l \mathbf{B}_i^l \mathbf{S}_i^l (\mathbf{B}_i^l)^T \tag{5}$$

where $\mathbf{S}_i^l$ denotes the scale matrix. The parameters in the trust regions are retrained with the scaled weight projection and the learnt scale matrices are stored in the memory for the inference phase. Particularly, during the forward transfer, the parameters are modified with the scale matrices as:

$$
\begin{aligned}
W_t^l &= \text{Proj}_{(U_i^l)^\perp}(W_t^l) + \text{Proj}_{U_i^l}^{S_i^l}(W_t^l) \\
&= W_t^l - \text{Proj}_{U_i^l}(W_t^l) + \text{Proj}_{U_i^l}^{S_i^l}(W_t^l)
\end{aligned}
\tag{6}
$$

where $(\cdot)^\perp$ denotes the orthogonal complemented subspace. However, as tasks come, increasing extra parameters are introduced by storing the scaling matrices. Our experiments demonstrate that TRGP requires around 5000% amount of the parameters regarding the network architecture after learning 20 tasks on MiniImageNet, see Figure 3-(c). Therefore, we propose our *Iterative Relaxing Gradient Projection* framework to facilitate forward knowledge transfer while maintaining a fixed network capacity by wisely reusing parameters within the frozen space.

## 3.2 RELAXING SUBSPACE SEARCHING

We first design a searching strategy to determine which part of the frozen space to relax based on the estimated importance characterized by the angle from the representation space spanned by current gradients $g_t^l$. The angle between a given space and a vector is defined in definition 3.1.

**Definition 3.1.** *(Angle between vector and space) We denote the angle between two inputs as $\Theta(\cdot)$ and the inner product between two vectors as $\langle \cdot \rangle$. The angle between a vector $v \in \mathbb{R}^n$ and a space $U^{n \times c} \subset \mathbb{R}^n$ is defined as the minimum angle between the given vector $v$ and any unit vector $u \in U$:*

$$\Theta(v, U) = \arccos \max_{u \in U} \frac{\langle v, u \rangle}{\|v\|} \tag{7}$$

Moreover, given the threshold $\gamma_t^l$, we define that a vector $d$ is *relaxable* when:

$$\Theta(d, R_{g,t}^l) \leq \gamma_t^l \tag{8}$$

where $R_{g,t}^l$ is constructed by compressing $g_t^l$ with Equation (3). For task $t$, we aim to find the relaxing subspace $V_t^l \subseteq U_{t-1}^l$ spanned all by *relaxable* vectors from $U_{t-1}^l$. Particularly, we implement with the modulus of the projection, namely:

$$
\begin{cases}
\min_{v \in V_t^l} \|\text{Proj}_{R_{g,t}^l}(v)\|_F \geq \zeta_t^l \|v\|_F \\
\max_{u \in U_{t-1}^{l,c}} \|\text{Proj}_{R_{g,t}^l}(u)\|_F < \zeta_t^l \|u\|_F
\end{cases}
\tag{9}
$$

where $\zeta_t^l = \cos \gamma_t^l$ is the threshold and $U_{t-1}^{l,c} = U_{t-1}^l \backslash V_t^l$ denotes the complemented subspace of $V_t^l$ with respect to $U_{t-1}^l$. Above criterion guarantees $\max_{u \in V_t^l} \Theta(u, R_{g,t}^l) \leq \gamma_t^l$ and $\min_{v \in U_{t-1}^{l,c}} \Theta(v, R_{g,t}^l) > \gamma_t^l$.

However, it is hard to construct $V_t^l$ directly from $U_{t-1}^l$. Therefore, we propose a simple yet efficient strategy to find the relaxing subspaces. With $V_t^l$ initiated as $\emptyset$, we select the closest vector to $R_{g,t}^l$ within $U_{t-1}^{l,c}$ by $\arg\min_{d \in U_{t-1}^{l,c}} \Theta(d, R_{g,t}^l)$. The selected vector $d$ is then appended into $V_t^l$ as basis if it satisfies criterion (8). We repeat this procedure until no *relaxable* vector left to get the target $V_t^l$. The pseudo-code of our searching strategy is provided in Algorithm 1.

Considering the scope of our procedure, we further provide theoretical analysis on the upper bound of the dimension of the selected subspace $V_t^l$, which is also the number of iterations. Here we introduce Lemma 3.2 and Theorem 3.3, which guarantee that the dimension of $V_T^l$ is no more than of the representation matrix. Detailed proof is provided in Appendix A.1 and A.2.

---

**Algorithm 1** Relaxing Subspace Searching

---

**Input:** gradient $\{g_t^l\}_{l=1}^L$, frozen subspace $\{U_{t-1}^l\}_{l=1}^L$ and thresholds $\{\epsilon_{th}^l, \gamma_t^l\}_{l=1}^L$
**Output:** relaxing subspace $\{V_t^l\}_{l=1}^L$
 1: **for** $l \in 1, ..., L$ **do**
 2:     Construct the significant representation space $R_{g,t}^l$ from gradients $g_t^l$ by Equation (3).
 3:     $V_t^l \leftarrow \emptyset$
 4:     **repeat**
 5:         $d \leftarrow \underset{d \in U_{t-1}^{l,c}}{\arg\min} \Theta(d, R_{g,t}^l)$
 6:         **if** $\Theta(d, R_{g,t}^l) \leq \gamma_t^l$ **then**
 7:             $V_t^l \leftarrow V_t^l \cup d$
 8:             $U_{t-1}^{l,c} \leftarrow U_{t-1}^l \backslash V_t^l$
 9:         **end if**
10:     **until** $\Theta(d, R_{g,t}^l) > \gamma_t^l$
11: **end for**

---

**Lemma 3.2.** *Denote the relaxed subspace as $V = \text{span}\{v_1, v_2, ..., v_N\}$, where $v_N$ is the last base included in $V$. Given representation subspace $U$, $\forall v \in V$, we have $\Theta(v, U) \leq \Theta(v_t, U)$.*

**Theorem 3.3.** *Denote $k_p$ as the dimension of the representation subspace and $k_l$ as the dimension of the relaxed subspace, the upper bound of $k_l$ is $k_p$, regardless of the frozen subspace.*

Moreover, according to Theorem 3.4, we figure that our strategy guarantees to find the maximum space within the whole solution set satisfying criterion (9), which further substantiates the efficiency of our searching strategy. We also include the corresponding proof in Appendix A.3.

**Theorem 3.4.** *The relaxed subspace obtained by Algorithm 1 takes up the maximum subspace of the whole solution set.*

To further validate our searching strategy, we propose IRGP-Exp, a modified version of our proposed IRGP, directly storing the parameters in the relaxed subspaces. For task $t$, we retrieve the corresponding relaxed subspaces $\{V_t^l\}_{l=1}^L$ and the scale matrices $\{S_t^l\}_{l=1}^L$ during the inference phase similar to TRGP. The modified parameters $W_{t,I}^l$ used for inference on task $t$ is:

$$W_{t,I}^l = W^l - \text{Proj}_{V_t^l}(W^l) + \text{Proj}_{V_t^l}^{S_t^l}(W^l) \qquad (10)$$

where $W^l$ denotes the parameters of layer $l$ of current network. Replacing the parameters in the relaxed subspaces with the parameters optimized in task $t$, the model achieves better performance.

### 3.3 ITERATIVE MODIFYING THE SCALE MATRIX WITH CONSTRAINTS

After getting the relaxed subspaces, we want to retrain inside parameters while consolidating previous knowledge, to facilitate forward knowledge transfer within a fixed network capacity. One direct way is to fine-tune the parameters with regularization such as EWC (Kirkpatrick et al., 2017). However, regularization terms are designed for explicit parameters, which are not applicable for implicit subspace in our framework. Therefore, we introduce the scaled weight projection (Lin et al., 2022) to modify explicit parameters instead. With scaled weight projection, we fine-tune the parameters within $\mathcal{V}_t^l$ by adding regularization term on scaling matrices $\boldsymbol{\mathcal{S}_t} = \{S_t^l\}_{l=1}^L$ instead of direct on target parameters. Specifically, the objective function of task $t$ is:

$$\mathcal{L}_t = \mathcal{L}(\mathcal{W}_t, \mathcal{D}^{(t)}) + \Sigma_{l=1}^L \beta_l \|S_t^l - \mathbb{1}(S_t^l)\|_2^2 \qquad (11)$$

where $\mathbb{1}(\cdot)$ denotes the identity matrix with the size of the rank of the input matrix and $\beta_l$ is the weight of the regularization term for layer $l$. During back propagation, gradients within frozen space $U_{t-1}^l$ are eliminated and parameters within $V_t^l$ are modified by $S_t^l$. Generally, in our *Iterative Relaxing Gradient Projection* framework, we adopt our searching strategy to determine the relaxed subspaces and modify those parameters with constraints on the scaling matrices.

However, during the training phase, the direction of gradients shifts sharply and frequently due to the steep learning scope of deep neural networks. Diverse subspaces would be selected in different training phases. Therefore, we iteratively execute Algorithm 1 during training until no extra subspace is required. Particularly, for each task, our model is optimized for limited epochs first. Then

we search for the target relaxing subspace and examine whether there exists a new subspace within the remaining frozen space. If extra frozen subspace is released, the scale of the scaling matrix requires to be modified, while TRGP maintains a fixed-size scaling matrix throughout training. Thus, to accommodate the increasing relaxing subspace, we propose to expand the scaling matrix with the identity matrix of the corresponding size as:

$$\mathbf{S}_t^{l,new} = \begin{pmatrix} \mathbf{S}_t^{l,old} & 0 \\ 0 & \mathbb{1}(V_t^{l,new}) \end{pmatrix} \tag{12}$$

where $V_t^{l,new}$ denotes the newly included relaxing subspace. If there is no extra relaxing subspace, we optimize our model thoroughly on current task. After training, the parameters within $V_t^l$ are further consolidated by Equation (6) and the scaling matrices are emptied to be identical matrices. The pseudo-code of our *Iterative Relaxing Gradient Projection* framework is provided in Appendix D.

## 4 EXPERIMENTS

### 4.1 EXPERIMENTAL SETUP

**Datasets:** We evaluate our framework on five datasets. Following Saha et al. (2021), we conduct experiments on **CIFAR-100 Split** (Krizhevsky & Hinton, 2009), **MiniImageNet** (Vinyals et al., 2016), **Permuted MNIST (MNIST)** (Kirkpatrick et al., 2017) and **CIFAR-100 Sup** (Yoon et al., 2019). Moreover, Serra et al. (2018) first propose **Mixture**, consists of CIFAR-10 (Krizhevsky & Hinton, 2009), MNIST (LeCun et al., 1998), CIFAR-100 (Krizhevsky & Hinton, 2009), SVHN (Netzer et al., 2011), FashionMNIST (Xiao et al., 2017), TrafficSigns (Stallkamp et al., 2011), FaceScrub (Ng & Winkler, 2014) and NotMNIST (Bulatov, 2011). Here we evaluate our framework on **Mixture** with seven tasks as a sequence except TrafficSigns[1]. Details and statistics of the datasets can be found in Appendix B.1. Moreover, we include the details of network architectures in Appendix B.2.

**Baselines:** We compare our approach with competitive and well-established approaches maintaining a fixed network capacity following Saha et al. (2021). We adopt ER_Res (Chaudhry et al., 2019) and A-GEM (Chaudhry et al., 2018) as representative replay-based methods: the memory buffer size for PMNIST, CIFAR-100 Split, MiniImageNet, and Mixture are 1000, 2000, 500 and 3000, respectively. For gradient projection approaches, we consider OWM (Zeng et al., 2019) and GPM (Saha et al., 2021). For regularization approaches, we compare against EWC (Kirkpatrick et al., 2017) and state-of-the-art HAT (Serra et al., 2018). We also include the "multitask" baseline jointly training all tasks in a single network, which is always considered as an upper bound for continual learning. Other implementation details are listed in Appendix B.3. We exclude expansion-based methods in the main experiments as they use continually growing architecture, which is out of the scope of our work.

**Metrics:** We first employ two standard evaluation metrics: Average Accuracy (ACC) (Mirzadeh et al., 2020) and Backward Transfer (BWT) (Lopez-Paz & Ranzato, 2017). Denote $A_{i,j}$ as the test accuracy of task $j$ after learning task $i$. ACC is the average test accuracy evaluated after learning all tasks, defined as $\text{ACC} = \frac{1}{T}\Sigma_{i=1}^{T}A_{T,i}$. BWT is the average accuracy decrease after learning following tasks, defined as $\text{BWT} = \frac{1}{T-1}\Sigma_{i=1}^{T-1}(A_{T,i} - A_{i,i})$. To evaluate forward knowledge transfer, we further introduce Forward Transfer (FWT) (Lopez-Paz & Ranzato, 2017) and $\Omega_{new}$ (Kemker et al., 2018). FWT reflects the influence of the observed tasks on new tasks in a zero-shot manner, while $\Omega_{new}$ indicates the capability of acquiring new tasks. The detailed definitions are provided in Appendix B.4. In this paper, we mainly focus on $\Omega_{new}$ among the three metrics and results on FWT are provided as well. Generally, the larger ACC, the better the approach. Forward and backward knowledge transfer evaluate the capability of learning and memorizing respectively.

### 4.2 MAIN RESULTS

We show the comparative results on four benchmarks in Table 1. The experiments on Mixture are implemented by us, while other results are reported from (Saha et al., 2021). We run each experiment five times and report the mean results. We include implementation details in Appendix B.3 and detailed results including forward transfer can be found in Appendix C.2. As shown in Table 1, our approach obtain the best accuracy with comparable forgetting across all datasets.

---

[1]We fail to access the TrafficSigns datasets as the links provided in (Stallkamp et al., 2011; Serra et al., 2018; Saha et al., 2021) are all expired

Table 1: Comparison of average accuracy and forgetting tested after learning all tasks. *Multitask* is under non-incremental setting. All results reported are averaged over 5 runs.

| Method | CIFAR-100 Split | | MiniImageNet | | PMNIST | | Mixture | |
|---|---|---|---|---|---|---|---|---|
| | ACC (%) | BWT (%) | ACC (%) | BWT (%) | ACC (%) | BWT (%) | ACC (%) | BWT (%) |
| Multitask* | 79.58 ± 0.54 | - | 69.46 ± 0.62 | - | 96.70 ± 0.02 | - | 81.29 ± 0.23 | - |
| OWM | 50.94 ± 0.60 | -30 ± 1 | - | - | 90.71 ± 0.11 | **-1 ± 0** | OOM | - |
| EWC | 68.80 ± 0.88 | -2 ± 1 | 52.01 ± 2.53 | -12 ± 3 | 89.97 ± 0.57 | -4 ± 1 | 69.62 ± 2.69 | -6 ± 4 |
| HAT | 72.06 ± 0.50 | **0 ± 0** | 59.78 ± 0.57 | -3 ± 0 | - | - | 77.54 ± 0.18 | **-1 ± 0** |
| A-GEM | 63.98 ± 1.22 | -15 ± 2 | 57.24 ± 0.72 | -12 ± 1 | 83.56 ± 0.16 | -14 ± 1 | 59.86 ± 1.01 | -29 ± 1 |
| ER_Res | 71.73 ± 0.63 | -6 ± 1 | 58.94 ± 0.85 | -7 ± 1 | 87.24 ± 0.53 | -11 ± 1 | 75.07 ± 0.55 | -12 ± 1 |
| GPM | 72.48 ± 0.40 | -1 ± 0 | 60.41 ± 0.61 | **-1 ± 0** | 93.91 ± 0.16 | -3 ± 0 | 77.49 ± 0.68 | -5 ± 0 |
| Ours (IRGP) | **73.52 ± 0.45** | -1 ± 0 | **61.26 ± 1.68** | -2 ± 1 | **94.20 ± 0.11** | -2 ± 0 | **77.91 ± 0.45** | -4 ± 0 |

Figure 2: Results on CIFAR-100 Split setting: (a) averaged accuracy after learning each task; (b) accuracy evolution of a randomly selected task; (c) accuracy tested on task $i$ after learning task $i$.

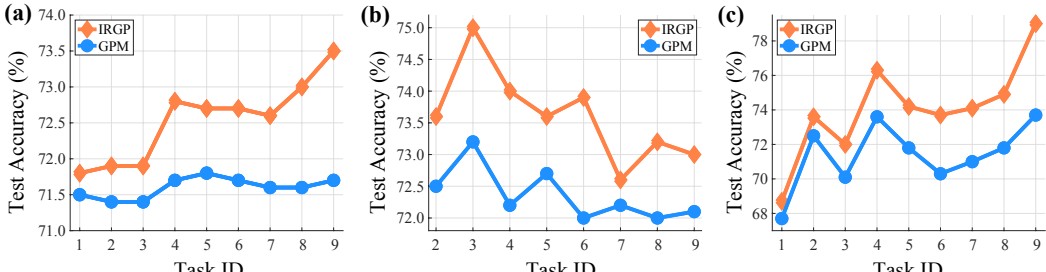

Compared with replay-based methods A-GEM and ER_Res, IRGP achieves at least around 2% higher ACC with less forgetting. For regularization-based methods, IRGP significantly dominates EWC across all benchmarks and outperforms HAT on MiniImageNet and PMNIST. Although HAT obtains less forgetting on the other two datasets, IRGP gains 1% better ACC on average. For gradient projection methods, we observe that IRGP achieves around 1% higher ACC on CIFAR-100 Split and MiniImageNet than GPM with comparable forgetting. On PMNIST and Mixture, IRGP improves the accuracy with less forgetting, reducing 1% BWT than GPM. The averaged accuracy after learning each task on CIFAR-100 Split exhibited in Figure 2-(a) further validates that IRGP universally outperforms GPM. We include the detailed results on other benchmarks in Appendix C.1.

Moreover, we compare the accuracy evolution of specific tasks during sequential tasks with GPM, which achieves the highest accuracy among selected baselines according to Table 1. Here we show the results of the second task on CIFAR-100 Split in Figure 2-(b). We further present the results of three randomly selected tasks on all benchmarks in Appendix C.3. We notice that IRGP achieves better accuracy right after learning a new task, in other words, gains better $\Omega_{new}$, which is also the purpose of our relaxing strategy. Without forward knowledge transfer, approaches may have limited performance even with less forgetting (Lopez-Paz & Ranzato, 2017).

Thus, we observe the accuracy tested after learning each task. As shown in Figure 2-(c), our approach achieves 2.7% better average accuracy on CIFAR-100 Split setting than GPM. Results on other benchmarks provided in Appendix C.2 further substantiate this phenomenon. As tasks keep coming, accumulated frozen spaces lead to decreasing optimization spaces for GPM. In the contrast, IRGP explores larger optimization spaces by relaxing previous frozen spaces. Thus, IRGP achieves better forward knowledge transfer by implicitly reusing the weights within the relaxed subspaces.

In brief, our approach universally outperforms selected baselines on all datasets in a fixed capacity. With comparable forgetting, IRGP achieves better forward knowledge transfer with larger optimization spaces against GPM. To validate the efficiency of our relaxing strategy, we further compare IRGP-Exp with well-established and competitive expansion-based methods in the next section.

### 4.3 COMPARED WITH EXPANSION-BASED METHODS

The above experiments exhibit the outstanding performance of our approach maintaining a fixed network capacity. By allocating new neurons or modules, expansion-based methods significantly mitigate backward interference with increasing capacity. Thus, to further validate our strategy, we compare IRGP and IRGP-Exp with relative expansion-based methods in this section.

Table 2: Results of ACC (%) and Capacity on CIFAR-100 Sup setting. *STL* is under non-incremental setting. All baselines are expansion-based methods except GPM.

| Metric | Methods | | | | | | |
|---|---|---|---|---|---|---|---|
| | STL* | PNN | DEN | RCL | APD | GPM | IRGP |
| ACC (%) | 61.00 | 50.76 | 51.10 | 51.99 | 56.81 | 57.72 | **58.12** |
| Capacity | 20.00 | 2.71 | 1.91 | 1.84 | 1.30 | 1.00 | 1.00 |

Table 3: L: compare IRGP-Exp with TRGP under an expansion setting. R: compare IRGP with TRGP within a fixed network capacity. We reports the results as (ACC / $\Omega_{new}$) for each experimental setting. Detailed results are provided in Tabel 16 and 17.

| Methods | Expansion | | | | Non-Expansion | | | |
|---|---|---|---|---|---|---|---|---|
| | IRGP-Exp | | T% | TRGP | TRGP-Reg | | | IRGP |
| | 50% | 80% | | | $w=1$ | $w=5$ | $w=50$ | |
| CIFAR | 75.15/74.76 | **75.38/75.02** | 75.06/74.91 | 74.46/75.01 | 71.85/72.87 | 72.08/73.12 | 72.46/72.91 | **73.52/74.78** |
| PMNIST | 96.68/97.18 | 96.99/**97.29** | **97.03**/97.26 | 96.34/97.23 | 73.69/95.20 | 71.51/95.60 | 72.43/95.80 | **94.20/96.19** |
| MiniImageNet | 60.81/62.35 | **62.03**/62.44 | 60.84/62.17 | 61.78/**63.29** | 55.81/60.28 | 58.81/62.77 | 22.69/20.12 | **61.26/62.80** |
| Mixture | 82.45/84.12 | 83.22/84.60 | **83.62**/83.97 | 83.54/**84.88** | 73.31/**84.05** | 74.71/83.48 | 17.36/7.22 | **77.91**/82.21 |

Following Saha et al. (2021), we perform experiments on CIFAR-100 Sup (Yoon et al., 2019). ACC results shown in Table 2 are averaged over 5 different sequence orders proposed by Yoon et al. (2017). We refer to the results of baselines from Saha et al. (2021). Capacity denotes the model capacity normalized with respect to the network used in GPM. Here we use the same model as GPM. According to Table 2, IRGP outperforms all baselines including GPM with the smallest capactiy.

Lin et al. (2022) proposed TRGP to expand the limited optimization spaces by retraining parameters within the selected trust regions, achieving superior performance. During the inference phase, TRGP reuses the parameters in corresponding trust regions memorized after learning this task. In contrast, GPM and our IRGP only store the representation of the frozen space. Therefore, although indeed a stable network capacity is allocated for each task, the entire memory size of TRGP grows continually. As shown in Figure 3-(c), after learning the last task on MiniImageNet setting, TRGP requires around 5000% extra parameters with respect to the network capacity. Results on other benchmarks provided in Appendix C.5 further substantiate that TRGP introduces a significant number of extra parameters. Thus, we categorize TRGP as an expansion-based method here.

As mentioned in Section 3.2, we propose IRGP-Exp to further validate our searching strategy. In this setting, the main difference between IRGP-Exp and TRGP is the strategy of deciding which part of the frozen space to reuse. We conduct experiments on all four benchmarks against TRGP. The results of ACC are provided in the left of Table 3. The percentages indicate the ratios of the rank of relaxing subspaces with respect to the corresponding frozen space. We evaluate three constant ratios and further use the ratios in TRGP, denoted as T%. According to the left of Table 3, IRGP-Exp already outperforms TRGP with relaxing only 50% of the frozen spaces on CIFAR100-Split and PMNIST. As TRGP selects the top 2 tasks as the trust regions, T% is larger than 80% most times. Moreover, our approach gains better ACC on all benchmarks with a comparable size of relaxing subspaces, which substantiate the efficiency of our subspace searching strategy.

We further modify TRGP as TRGP-Reg with similar regularization terms on the scale matrices as our IRGP to compare the relaxing strategies. We report the results on four benchmarks with three representative regularization weights $w$ on TRGP-Reg in the right of Table 3. As shown in Table 3, IRGP significantly outperforms TRGP-Reg, especially on PMNIST, gaining over 20% ACC improvement. Generally, IRGP achieves better or comparable $\Omega_{new}$ than TRGP under or without the constraint of a fixed network capacity. Detailed results are included in Appendix C.7.

## 5 ANALYSIS AND DISCUSSION

To gain a deeper insight into IRGP, we investigate the trend of scales of the subspaces relaxed by our strategy. With the theoretical upper bound of the rank of the relaxed subspace provided in Theorem 3.3, we further inspect the ratios of the relaxed subspaces concerning corresponding frozen spaces in practical. Results of the last layer on three different settings are provided in Figure 3-(a). As shown in Figure 3-(a), relaxing ratios maintain a stable trend, fluctuating smoothly in a similar

Figure 3: (a) Relaxing ratios of the last layer on CIFAR-100 Split, MiniImageNet, and PMNIST. (b) Test accuracy of GPM and IRGP of different $\epsilon$ on CIFAR-100 Split. The optimum value of GPM is annotated by a red circle. (c) Ratios of the amount of extra parameters concerning the amount of the parameters of the network architecture on MiniImageNet.

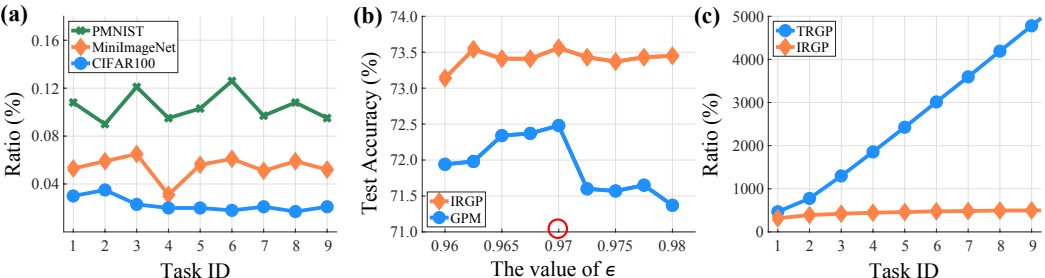

range over sequential tasks, on both benchmarks. As different tasks explore different optimization directions, ideal relaxing subspaces vary across tasks, in accord with the fluctuation of our results.

The dimension of the frozen spaces keeps growing as the tasks come, leading to expanding range for searching relaxing subspaces. Therefore, the computation complexity and time consumption are supposed to increase gradually. To investigate the practical efficiency of our approach, we further reported the time consumption of IRGP on CIFAR-100 Split and MiniImageNet against other baselines in Appendix C.4. According to Table 14, our approach takes around 60% more time than GPM on both settings. In general, the practical efficiency of our approach is accecptable.

To understand our relaxing strategy better, we further conduct experiments on different thresholds $\epsilon$ mentioned in Equation (3), which regulate the criterion of the frozen spaces. Saha et al. (2021) argue that $\epsilon$ controls the scale of the frozen space to mediate the stability-plasticity dilemma, and thus is critical for GPM. However, IRGP enables the frozen space to be dynamically regulated regarding the current task. Therefore, $\epsilon$ plays a much less important role in IRGP. We present the performance of different $\epsilon$ on CIFAR-100 Split in Figure 3-(b). As shown in Figure 3-(b), the performance of GPM drops significantly when $\epsilon \geq 0.97$, the optimal value reported in GPM, while IRGP consistently performs well even with $\epsilon = 0.98$. Generally, IRGP is more robust on the threshold $\epsilon$.

In the contrast, IRGP mediates the stability-plasticity dilemma by controlling the dimension and flexibility of the relaxing space by $\zeta$ in Equation (9) and $\beta$ in Equation (11) respectively. We present the results on CIFAR-100 Split in Table 4, where $\zeta_{conv}$ denotes the hyper-parameter for convolutional layers and $\zeta_{fc}$ denotes the hyper-parameter for fully connected layers. As shown in Table 4, a larger $\zeta$ guarantees less forgetting, as a result of smaller relaxing subspaces. Detailed results are provided in Appendix C.6. Similarly, we observe less forgetting on larger $\beta$, which constrains the update of parameters within the relaxing subspace more strictly. However, strict constraints also lead to limited performance on new tasks as discussed in Section 4. Generally, $\zeta$ and $\beta$ work together to overcome catastrophic forgetting with better forward transfer.

Table 4: Ablation study of $\zeta = \cos\gamma$ and $\beta$ on CIFAR100-Split, where $\gamma$ is the threshold for the relaxing strategy and $\beta$ is the weight of the regularization terms.

| $\zeta_{conv}$ | $\zeta_{fc}$ | $\beta$ | ACC (%) | BWT (%) |
|---|---|---|---|---|
| 0.20 | 0.20 | 1.0 | 71.97 | -3.0 |
| 0.50 | 0.50 | 1.0 | 72.58 | -2.3 |
| 0.80 | 0.80 | 1.0 | 73.34 | -2.2 |
| 0.90 | 0.90 | 1.0 | 73.28 | -1.2 |
| 0.95 | 0.90 | 0.0 | 73.15 | -2.0 |
| 0.95 | 0.90 | 0.1 | 73.32 | -1.3 |
| **0.95** | **0.90** | **1.0** | **73.52** | **-0.9** |
| 0.95 | 0.90 | 5.0 | 73.14 | -0.7 |
| 0.95 | 0.90 | 10.0 | 72.81 | -0.6 |
| 0.95 | 0.95 | 1.0 | 73.08 | -0.6 |

## 6 CONCLUSION

In this paper, we propose a novel continual learning approach that facilitates the forward knowledge transfer in gradient projection methods with a fixed network capacity by iteratively searching and relaxing subspaces within the frozen space to expand the optimization space. Extensive experiments demonstrate that our IRGP framework surpasses related state-of-the-art approaches on diverse benchmarks. Moreover, we propose a modified version expanding the architecture with relaxing subspaces, achieving better average accuracy than other expansion-based methods. We further provide solid proof and analysis validating the efficiency of our algorithm.

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

## A  PROOF

### A.1  PROOF OF LEMMA 3.2

For simplification, all vectors here are assumed to be unit vectors, namely $\|v\| = 1$. Depicting the angle by projection form, Lemma 3.2 can be expressed as:

$$\forall v \in V, \|\mathrm{Proj}_U(v)\| \geq \|\mathrm{Proj}_U(v_t)\| \tag{13}$$

Denote $\mathbf{B} = [u_1, ..., u_m]$ as the representation matrix of the representation subspace $U = \mathrm{span}\{u_1, ..., u_m\}$, where $u_i$ is the $i$-th normalized base of $U$. Lemma 3.2 can be expressed as:

$$\forall v \in V, v^T \mathbf{B}\mathbf{B}^T v \geq v_t^T \mathbf{B}\mathbf{B}^T v_t \tag{14}$$

As $v_i$s are the basis iteratively appended by Algorithm 1, for any $i \leq j$, we have:

$$v_i^T \mathbf{B}\mathbf{B}^T v_i \geq v_j^T \mathbf{B}\mathbf{B}^T v_j \tag{15}$$

Therefore, to prove Lemma 3.2, it suffices to prove the following lemma.

If there exists a vector $v = \Sigma_{i=1}^{t+1} w_i v_i$ satisfying the condition $v^T \mathbf{B}\mathbf{B}^T v < v_t^T \mathbf{B}\mathbf{B}^T v_t$, then we can find another vector $v' = \Sigma_{i=k}^{t+1} w_i' v_i$ such that $v'^T \mathbf{B}\mathbf{B}^T v' > v_k^T \mathbf{B}\mathbf{B}^T v_k$, which contradicts Algorithm 1 where $\Theta(v_k, U) \le \Theta(v, U)$ for $\forall v \in \mathrm{span}\{v_k, ..., v_{t+1}\}$.

First, we consider a special case where the current relaxed subspace has only one base, denoted by $V = \mathrm{span}\{v_1\}$. Assume there exists $v = w_1 v_1 + w_2 v_2$ that $v^T \mathbf{B}\mathbf{B}^T v < v_2^T \mathbf{B}\mathbf{B}^T v_2$, we have:

$$w_1^2 v_2^T \mathbf{B}\mathbf{B}^T v_2 > w_1^2 v_1^T \mathbf{B}\mathbf{B}^T v_1 + 2 w_1 w_2 v_1^T \mathbf{B}\mathbf{B}^T v_2 \tag{16}$$

Construct $v' = w_2 v_1 - w_1 v_2$, we have:

$$\begin{aligned} v'^T \mathbf{B}\mathbf{B}^T v' &= w_2^2 v_1^T \mathbf{B}\mathbf{B}^T v_1 + w_1^2 v_2^T \mathbf{B}\mathbf{B}^T v_2 - 2 w_1 w_2 v_1^T \mathbf{B}\mathbf{B}^T v_2 \\ &> w_2^2 v_1^T \mathbf{B}\mathbf{B}^T v_1 + w_1^2 v_2^T \mathbf{B}\mathbf{B}^T v_2 + w_1^2 v_1^T \mathbf{B}\mathbf{B}^T v_1 - w_1^2 v_2^T \mathbf{B}\mathbf{B}^T v_2 \\ &= v_1^T \mathbf{B}\mathbf{B}^T v_1 \end{aligned} \tag{17}$$

which contradicts $\Theta(v_1, U) \le \Theta(v, U)$ for $\forall v \in \mathrm{span}\{v_1, v_2\}$.

Then we consider the general case $V = \mathrm{span}\{v_1, ..., v_t\}$. For $\forall v \in V$, we have $\Theta(v, U) \le \Theta(v_t, U)$. After $v_t$ is included, we assume that there exists $v \in V$ that $\Theta(v, U) > \Theta(v_{t+1}, U)$. Then we can find the minimum $s$ satisfying that there exists $v = \Sigma_{i=1}^{s} w_i v_i + w_{t+1} v_{t+1}$ that $\Theta(v, U) > \Theta(v_{t+1}, U)$ and for $\forall v' = \Sigma_{i=1}^{s-1} w_i v_i + w_{t+1} v_{t+1}$, we have $\Theta(v', U) \le \Theta(v_{t+1}, U)$. When $s = 1$, it is similar to the special case, so the proof is omitted. Thus, we consider the case where $s \ge 2$. For simplification, we express $v$ as $v = c_0 v_0 + c_1 v_s + c_2 v_{t+1}$ with $v_0 = \Sigma_{i=1}^{s-1} a_i v_i$, where $c_i$s and $a_i$s are coefficients. We have:

$$(c_0 v_0 + c_1 v_s + c_2 v_{t+1})^T \mathbf{B}\mathbf{B}^T (c_0 v_0 + c_1 v_s + c_2 v_{t+1}) < v_{t+1}^T \mathbf{B}\mathbf{B}^T v_{t+1} \tag{18}$$

As $\Theta(w_1 v_0 + w_2 v_s, U) \le \Theta(v_s, U) \le \Theta(v_{t+1}, U)$, we have:

$$(w_1 v_0 + w_2 v_s)^T \mathbf{B}\mathbf{B}^T (w_1 v_0 + w_2 v_s) \ge v_s^T \mathbf{B}\mathbf{B}^T v_s \tag{19}$$

which is:

$$w_1^2 v_0^T \mathbf{B}\mathbf{B}^T v_0 + 2 w_1 w_2 v_0^T \mathbf{B}\mathbf{B}^T v_s \ge w_1^2 v_s^T \mathbf{B}\mathbf{B}^T v_s \ge w_1^2 v_t^T \mathbf{B}\mathbf{B}^T v_t \tag{20}$$

Similarly we have:

$$w_1^2 v_0^T \mathbf{B}\mathbf{B}^T v_0 + 2 w_1 w_2 v_0^T \mathbf{B}\mathbf{B}^T v_t \ge w_1^2 v_t^T \mathbf{B}\mathbf{B}^T v_t \tag{21}$$

Then we can express Equation (18) as:

$$v_{t+1}^T \mathbf{B}\mathbf{B}^T v_{t+1} > (c_0^2 + c_2^2) v_t^T \mathbf{B}\mathbf{B}^T v_t + c_1^2 v_s^T \mathbf{B}\mathbf{B}^T v_s + 2 c_1 c_2 v_s^T \mathbf{B}\mathbf{B}^T v_t \tag{22}$$

As $\|v\| = \|v_i\| = 1$, $c_0^2 + c_1^2 + c_2^2 = 1$. Then we have:

$$- 2 c_1 c_2 v_s^T \mathbf{B}\mathbf{B}^T v_t > c_1^2 v_s^T \mathbf{B}\mathbf{B}^T v_s - c_1^2 v_{t+1}^T \mathbf{B}\mathbf{B}^T v_{t+1} \tag{23}$$

Construct $v' = \frac{c_2 v_s - c_1 v_{t+1}}{\sqrt{c_1^2 + c_2^2}}$, we have:

$$\begin{aligned} v'^T \mathbf{B}\mathbf{B}^T v' &= \frac{1}{c_1^2 + c_2^2} (c_2^2 v_s^T \mathbf{B}\mathbf{B}^T v_s + c_1^2 v_{t+1}^T \mathbf{B}\mathbf{B}^T v_{t+1} - 2 c_1 c_2 v_s^T \mathbf{B}\mathbf{B}^T v_t) \\ &> \frac{1}{c_1^2 + c_2^2} (c_2^2 v_s^T \mathbf{B}\mathbf{B}^T v_s + c_1^2 v_{t+1}^T \mathbf{B}\mathbf{B}^T v_{t+1} + c_1^2 v_s^T \mathbf{B}\mathbf{B}^T v_s - c_1^2 v_{t+1}^T \mathbf{B}\mathbf{B}^T v_{t+1}) \\ &= v_s^T \mathbf{B}\mathbf{B}^T v_s \end{aligned}$$

$$\tag{24}$$

which contradicts $\Theta(v_s, U) \leq \Theta(v, U)$ for $\forall v \in \mathrm{span}\{v_s, v_{s+1}, ..., v_t, v_{t+1}\}$.

Thus, for $\forall v \in V = \mathrm{span}\{v_1, ..., v_t\}$, we have $\Theta(v, U) \leq \Theta(v_t, U)$.

## A.2  PROOF OF THEOREM 3.3

Denote the relaxed subspace and the representation subspace as $V$ and $U = \mathrm{span}\{u_1, ..., u_{k_p}\}$ respectively. With Lemma 3.2, we have $\forall v \in V$, $\Theta(v, U) \leq \Theta(v_t, U) < \frac{\pi}{2}$. In other words,

$$\forall v \in V, \ v \not\perp U \tag{25}$$

For the sake of contradiction, assume the dimension of $V$ is larger than $k_p$, namely $\dim(V) > \dim(U) = k_p$. denote $U^c$ as the complemented subspace of $U$ with respect to the whole space $\mathbb{R}$. Obviously, $\dim(U^c) = n - k_p$, where $n$ is the dimension of $\mathbb{R}$. Then we have:

$$\begin{aligned}
\dim(V \cap U^c) &= \dim(V) + \dim(U^c) - \dim(V + U^c) \\
&> k_p + (n - k_p) - n = 0
\end{aligned} \tag{26}$$

Thus, there exists $v' \in V$ such that $v' \in U^c$ too. As $U^c$ is the complemented subspace, $\forall u \in U^c$, $u \perp U$. Then we have $v' \perp U$, which contradicts Equation (25). Therefore, the assumption is aborted. The upper bound of the dimension of $V$ is $k_p$, namely the dimension of the representation subspace $U$.

## A.3  PROOF OF THEOREM 3.4

Denote the whole solution set as $S = \{u | \Theta(u, U) \leq \gamma \text{ and } u \in U^f\}$ where $U$ is the representation subspace, $U^f$ is the frozen space and $\gamma$ is the threshold. Theorem 3.4 can be expressed as that all subspace $V' \subseteq S$ satisfies $\dim(V') \leq \dim(V)$, where $V$ is the relaxed subspace obtained by Algorithm 1. Similar to Theorem 3.3, we assume there exists $V' \subseteq S$ that $\dim(V') > \dim(V)$. denote $V_f^c$ as the complemented subspace of $V$ with respect to the frozen space $U^f$. According to Algorithm 1, for $\forall v \in V_f^c$, we have $\Theta(v, U) > \gamma$. We also have:

$$\dim(V' \cap V_f^c) = \dim(V') + \dim(V_f^c) - \dim(V' + V_f^c) > 0 \tag{27}$$

Thus, there exists $v' \in V'$ that $v' \in U_f^c$, namely there exists $v' \in S$ that $\Theta(v', U) > \gamma$, which is contradict. Therefore, the relaxed subspace obtained by Algorithm 1 takes up the maximum subspace of the whole solution set.

## B  EXPERIMENTAL SETUP

### B.1  DATASETS

Here we introduce the datasets we use for evaluation. **1) CIFAR-100 Split** Saha et al. (2021) constructed **CIFAR-100 Split**, by splitting CIFAR100 (Krizhevsky & Hinton, 2009) into 10 tasks where each task has 10 classes. **2) MiniImageNet** Following Saha et al. (2021), we split MiniImageNet (Vinyals et al., 2016) into 20 sequential tasks with 5 classes each. **3) Permuted MNIST (PMNIST)** PMNIST (Kirkpatrick et al., 2017) is a variant of MNIST (LeCun et al., 1998) where each task has a different permutation of inputting images, consists of 10 sequential tasks with 10 classes each. **4) CIFAR-100 Sup** Following Yoon et al. (2019), we adopt **CIFAR-100 Sup** consisting of 20 superclasses as sequential tasks. **5) Mixture** Serra et al. (2018) first proposed Mixture consisting of 8 datasets, including CIFAR-10 (Krizhevsky & Hinton, 2009), MNIST (LeCun et al., 1998), CIFAR-100 (Krizhevsky & Hinton, 2009), SVHN (Netzer et al., 2011), FashionM-NIST (Xiao et al., 2017), TrafficSigns (Stallkamp et al., 2011), FaceScrub (Ng & Winkler, 2014), and NotMNIST (Bulatov, 2011), from which Ebrahimi et al. (2020) further constructed 5-Datasets. Here we follow the original harder benchmark. Particularly, we consider all tasks as a sequence except TrafficSigns (Stallkamp et al., 2011), which we failed to access. Among all evaluated datasets,

PMNIST is a benchmark under the domain-incremental scenario, while other four datasets are under the task-incremental scenario.

Moreover, we provide the statistics of selected datasets in Table 5 and Table 6. For the Mixture benchmark, the images of MNIST, FashionMNIST, and notMNIST are replicated across all RGB channels following Serra et al. (2018).

Table 5: Statistics of CIFAR-100 Split, MiniImageNet, and PMNIST.

|  | CIFAR-100 Split | CIFAR-100 Sup | MiniImageNet | PMNIST |
|---|---|---|---|---|
| Image Size | $32 \times 32$ | $32 \times 32$ | $84 \times 84$ | $28 \times 28$ |
| Channels | 3 | 3 | 3 | 1 |
| Classes | 100 | 100 | 100 | 10 |
| Tasks | 10 | 20 | 20 | 10 |
| Classes/task | 10 | 5 | 5 | 10 |
| Training Samples/task | 4,750 | 2,375 | 2,375 | 54,000 |
| Validation Samples/task | 250 | 125 | 125 | 6,000 |
| Testing Samples/task | 1,000 | 500 | 500 | 10,000 |

Table 6: Statistics of Mixture benchmark.

| Dataset | Classes | # Taining | # Validation | # Testing |
|---|---|---|---|---|
| CIFAR-10 (Krizhevsky & Hinton, 2009) | 10 | 47,500 | 2,500 | 10,000 |
| MNIST (LeCun et al., 1998) | 10 | 57,000 | 3,000 | 10,000 |
| CIFAR-100 (Krizhevsky & Hinton, 2009) | 100 | 47,500 | 2,500 | 10,000 |
| SVHN (Netzer et al., 2011) | 10 | 69,595 | 3,662 | 26,032 |
| FashionMNIST (Xiao et al., 2017) | 10 | 57,000 | 3,000 | 10,000 |
| FaceScrub (Ng & Winkler, 2014) | 100 | 19,570 | 1,030 | 2,289 |
| NotMNIST (Bulatov, 2011) | 10 | 16,011 | 842 | 1,873 |

### B.2 MODEL DETAILS

**MLP architecture:** We adopt a 3-layer model including two hidden layers with 100 neurons each for the PMNIST setting, the same as Lopez-Paz & Ranzato (2017). ReLU is used as the activate function here and for all other architectures. Also, we use softmax with cross entropy loss on all settings.

**AlexNet architecture:** For CIFAR-100 Split setting, we adopt the same network as Serra et al. (2018) with batch normalization, including two fully connected layers and three convolutional layers. The convolutional layers have $4 \times 4$, $3 \times 3$, and $2 \times 2$ kernel sizes with 64, 128, and 256 filters respectively. After each convolutional layer, we add batch normalization and $2 \times 2$ max-pooling. Each fully connected layer has 2048 units. For the first two layers, we use the dropout of 0.2, and for the rest layers, we use the dropout of 0.5.

**Modified LeNet-5 architecture:** For the CIFAR-100 Sup setting, a modified LeNet-5 architecture consisting of two convolutional layers and two fully connected layers is adopted, similar to Saha et al. (2021). Max-pooling of $3 \times 2$ is used after each convolutional layer. The last two layers have 800 and 500 units respectively.

**Reduced ResNet-18 architecture:** We adopt the same reduced ResNet-18 architecture as Saha et al. (2021) for the MiniImageNet and Mixture settings, using $2 \times 2$ average-pooling before the classifier layer instead of the $4 \times 4$ average-pooling used by Lopez-Paz & Ranzato (2017). Moreover, we present the dimension of the representation space of each layer of our architectures in Table 7.

### B.3 IMPLEMENTATION DETAILS

We use the official implementation of GPM (Saha et al., 2021), OWM (Zeng et al., 2019), HAT Serra et al. (2018), and TRGP (Lin et al., 2022). Moreover, we implement A-GEM and ER_Res with the

Table 7: Dimension of the representation space of each layer.

| Network | Depth | Dimension of the representation space |
|---|---|---|
| MLP | 3 layers | 784; 100; 100 |
| AlexNet | 5 layers | 48; 576; 512; 1,024; 2,048 |
| LeNet-5 | 4 layers | 75; 500; 3,200; 800 |
| ResNet-18 | 17 layers and 3 short-cut connections | 27; 180; 180; 180; 180; 180; 360; 20; 360; 360; 360; 720; 40; 720; 720; 720; 1,440; 80; 1,440; 1,440 |

official implementation by Chaudhry et al. (2018) and implement EWC with the implementation by Serra et al. (2018). Following Saha et al. (2021) and Lin et al. (2022), we run all experiments five times on an established seed without fixing the cuda settings for a fair comparison. Particularly, we use five random seeds on PMNIST where there is no diversity on a single seed. For CIFAR-100 Sup, we use five different orders provided by Yoon et al. (2019). Following Saha et al. (2021), we report the experimental results of replay-base methods A-GEM and ER_Res on the Mixture dataset with the same buffer size as GPM and our IRGP, which is 8.98M in term of the number of parameters for the Resnet18 architecture.

On CIFAR-100 Split, MiniImageNet, and PMNIST, we follow the hyper-parameters utilized by Saha et al. (2021) and Lin et al. (2022), including learning rate, batch size, and the threshold $\epsilon$. On Mixture, as we adopt the same network architecture Saha et al. (2021) use on their 5-Dataset setting, we follow the provided learning rate and batch size as well. Moreover, for the threshold $\epsilon$ in GPM, we conduct experiments with $\epsilon$ in the range of $0.95$ to $1$ provided in (Saha et al., 2021) and report the best results whose $\epsilon$ is $0.955$. We further use $\epsilon = 0.96$ for all layers in our IRGP.

As discussed in Section 5, the threshold $\zeta = \cos \gamma$ controls the criterion of the relaxing subspace. For CIFAR100-Split and PMNIST, we use $\zeta = 0.95$ for convolutional layers and $\zeta = 0.9$ for fully connected layers. For MiniImageNet and Mixture, we use the same $\zeta$ for all layers, $0.95$ and $0.9$ respectively. Furthermore, we set the regularization weight as 5 for the ResNet18 architecture and 1 for others. Particularly, we run all the experiments on a single NVIDIA GeForce RTX 2080 Ti GPU.

### B.4 METRICS

Here we present the detailed definitions of the metrics evaluating the forward knowledge transfer. $\Omega_{new}$ (Kemker et al., 2018). Denote $b_i$ as the test accuracy of task $i$ at random initialization, FWT, first proposed by Lopez-Paz & Ranzato (2017), is defined as $FWT = \frac{1}{T-1}\Sigma_{i=2}^{T}(A_{i-1,i} - b_i)$, evaluating the zero-shot performance of the initialization with respect to the observed tasks. While $\Omega_{new}$, first proposed by Kemker et al. (2018), is defined as $\Omega_{new} = \frac{1}{T-1}\Sigma_{i=2}^{T}(A_{i,i} - b_i)$, reflecting the test accuracy on new tasks based on the learnt knowledge. As $b_i$ stays still across different approaches, we consider $\Omega_{new} = \frac{1}{T-1}\Sigma_{i=2}^{T}A_{i,i}$ for simplicity. For this simplified $\Omega_{new}$, we have: $\Omega_{new} = \frac{T}{T-1}ACC - BWT - \frac{1}{T-1}A_{1,1}$, with the ACC and BWT defined in Section 4.1.

## C EXPERIMENTAL RESULTS

### C.1 FINAL ACCURACY

We provide the test accuracy after learning each task on other benchmarks here. As discussed in Section 4, our IRGP universally outperforms GPM over the task sequence on all benchmarks.

### C.2 FORWARD KNOWLEDGE TRANSFER

We provide the detailed forward knowledge transfer performance on all four benchmarks here. First, we present the results of $\Omega_{new}$ and the detailed accuracy of each task after learning it in Table 8 to 11. According to Table 10 and Table 11, IRGP achieves a similar forward knowledge transfer compared with GPM. For other benchmarks, IRGP improves $\Omega_{new}$ by 2.7% and 1.8% on CIFAR-100 Split and MiniImageNet respectively, as shown in Table 8 and Table 9. Moreover, we provided the detailed results including the standard deviation and other baselines in Table 12. According to Table 12, our IRGP consistently achieves the best or second best forward knowledge transfer compared with all

Figure 4: Average accuracy after learning each task on (a) MiniImageNet, (b) PMNIST, and (c) Mixture.

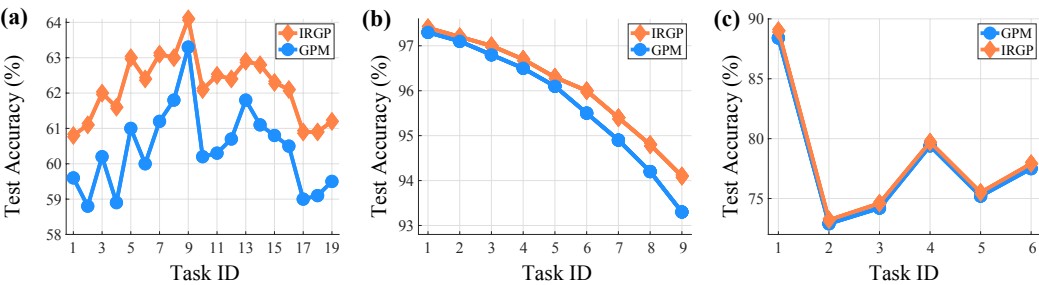

baselines without extra memory buffer. For CIFAR100-Split, IRGP even gains 2.7% better $\Omega_{new}$ than HAT, which is the second best baseline in this setting.

Table 8: The accuracy tested on task $i$ after learning task $i$ and $\Omega_{new}$ on CIFAR-100 Split.

| Method | 1 | 2 | 3 | 4 | 5 | 6 | 7 | 8 | 9 | Avg ($\Omega_{new}$) |
|---|---|---|---|---|---|---|---|---|---|---|
| GPM | 67.7 | 72.5 | 70.1 | 73.6 | 71.8 | 70.3 | 71.0 | 71.8 | 73.7 | 71.4 |
| IRGP | 68.7 | 73.6 | 72.0 | 76.3 | 74.2 | 73.7 | 74.1 | 74.9 | 79.0 | 74.1 |

Table 9: The accuracy tested on task $i$ after learning task $i$ and $\Omega_{new}$ on MiniImageNet.

| Method | 1 | 2 | 3 | 4 | 5 | 6 | 7 | 8 | 9 | 10 | 11 | 12 | 13 | 14 | 15 | 16 | 17 | 18 | 19 | Avg ($\Omega_{new}$) |
|---|---|---|---|---|---|---|---|---|---|---|---|---|---|---|---|---|---|---|---|---|
| GPM | 59.0 | 57.1 | 65.6 | 59.6 | 76.2 | 57.2 | 66.5 | 72.8 | 81.5 | 42.0 | 59.6 | 62.1 | 62.3 | 58.9 | 57.2 | 59.0 | 50.7 | 65.7 | 57.3 | 61.6 |
| IRGP | 57.6 | 60.1 | 66.0 | 62.2 | 77.0 | 62.5 | 65.7 | 73.6 | 82.3 | 44.4 | 64.0 | 59.2 | 64.5 | 60.0 | 59.6 | 60.5 | 50.0 | 68.3 | 55.6 | 62.8 |

Table 10: The accuracy tested on task $i$ after learning task $i$ and $\Omega_{new}$ on PMNIST.

| Method | 1 | 2 | 3 | 4 | 5 | 6 | 7 | 8 | 9 | Avg ($\Omega_{new}$) |
|---|---|---|---|---|---|---|---|---|---|---|
| GPM | 97.5 | 97.4 | 97.0 | 96.8 | 96.5 | 96.2 | 96.2 | 95.8 | 95.1 | 96.5 |
| IRGP | 97.4 | 97.2 | 97.0 | 96.5 | 96.3 | 96.1 | 95.7 | 94.9 | 94.8 | 96.2 |

Table 11: The accuracy tested on task $i$ after learning task $i$ and $\Omega_{new}$ on Mixture.

| Method | 1 | 2 | 3 | 4 | 5 | 6 | Avg ($\Omega_{new}$) |
|---|---|---|---|---|---|---|---|
| GPM | 99.0 | 43.7 | 87.3 | 99.1 | 69.9 | 93.4 | 82.1 |
| IRGP | 99.1 | 42.9 | 87.6 | 99.1 | 70.6 | 93.5 | 82.2 |

As mentioned in Section 4.1, final accuracy (ACC), forgetting (BWT) and forward knowledge transfer ($\Omega_{new}$) are jointly considered to evaluate a continual learner. Despite our relaxing strategy focusing on facilitating the forward knowledge transfer, IRGP gains better final accuracy with comparable forgetting over all benchmarks, according to Table 1. Generally speaking, IRGP achieves superior performance than previous baselines with a fixed network capacity.

Table 12: Comparison of forward knowledge transfer on four benchmarks, evaluated by $\Omega_{new}$.

| Datasets | A-GEM | ER_Res | EWC | HAT | GPM | IRGP |
|---|---|---|---|---|---|---|
| PMNIST | **97.4 ± 0.0** | 97.4 ± 0.5 | 93.1 ± 0.7 | - | **96.6 ± 0.0** | 96.5 ± 0.1 |
| CIFAR100-Split | **77.5 ± 0.4** | 77.1 ± 0.2 | 69.9 ± 1.1 | 71.5 ± 0.6 | 71.5 ± 0.5 | **74.1 ± 0.2** |
| MiniImageNet | 67.6 ± 1.2 | **69.5 ± 0.4** | 63.5 ± 2.8 | 62.6 ± 0.6 | 61.6 ± 0.6 | 62.8 ± 0.8 |
| Mixture | 85.9 ± 0.4 | **86.4 ± 0.2** | 74.4 ± 1.0 | 79.3 ± 0.2 | 82.1 ± 0.4 | **82.2 ± 0.4** |

Moreover, we provide the result of FWT (using the definition in (Lopez-Paz & Ranzato, 2017)) on all benchmarks in Table 13. According to Table 13, although our method facilitates the forward knowledge transfer reflected by $\Omega_{new}$, IRGP achieves better FWT than GPM on all three task-incremental benchmarks.

Table 13: Comparison of forward knowledge transfer between GPM and IRGP, evaluated by FWT.

| Methods | CIFAR-100 Split | | MiniImageNet | | PMNIST | | Mixture | |
|---|---|---|---|---|---|---|---|---|
| | FWT (%) | STD (%) | FWT (%) | STD (%) | FWT (%) | STD (%) | FWT (%) | STD (%) |
| GPM | -0.65 | **0.18** | -0.26 | 0.88 | **+0.66** | 1.17 | -0.52 | 1.49 |
| Ours (IRGP) | **+0.20** | 0.36 | **+0.52** | **0.35** | -0.63 | **0.97** | **+0.13** | **0.77** |

### C.3 ACCURACY EVOLUTION

Here we present the accuracy tested on three randomly selected tasks after learning them on four benchmarks. We select the 2nd, 4th, and 6th tasks for CIFAR-100 Split, MiniImageNet, and PM-NIST. As there are only 7 tasks in Mixture, we select the 1st, 3rd, and 5th tasks. Generally, IRGP outperforms GPM on selected tasks over the sequence. We further notice that the improvement is more significant on later tasks as a result of larger relaxing subspaces, as discussed in Section 5.

Figure 5: Accuracy evolution of the (a) 2nd, (b) 4th, and (c) 6th task on CIFAR-100 Split.

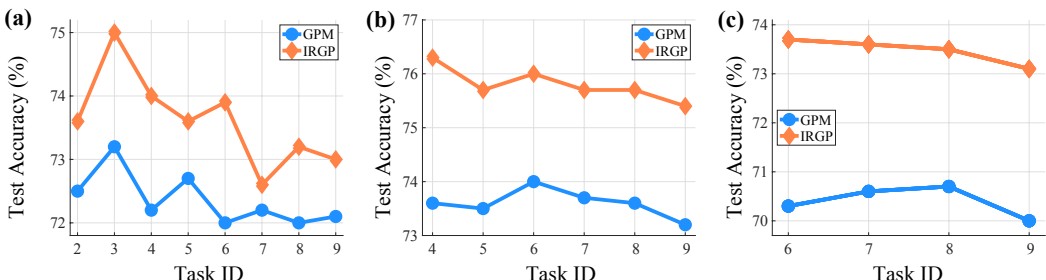

Figure 6: Accuracy evolution of the (a) 2nd, (b) 4th, and (c) 6th task on MiniImageNet.

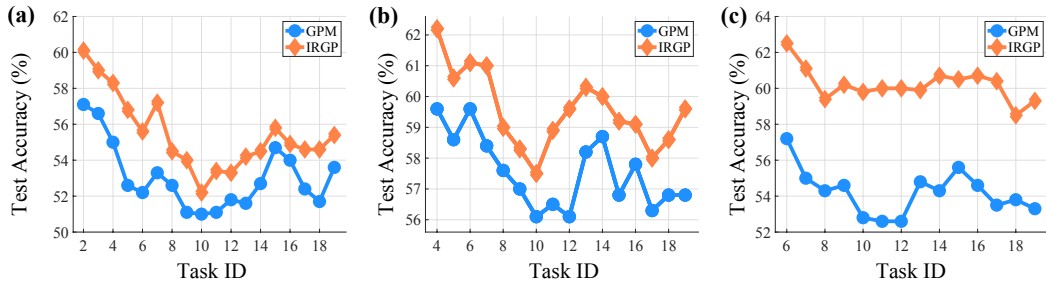

Figure 7: Accuracy evolution of the (a) 2nd, (b) 4th, and (c) 6th task on PMNIST.

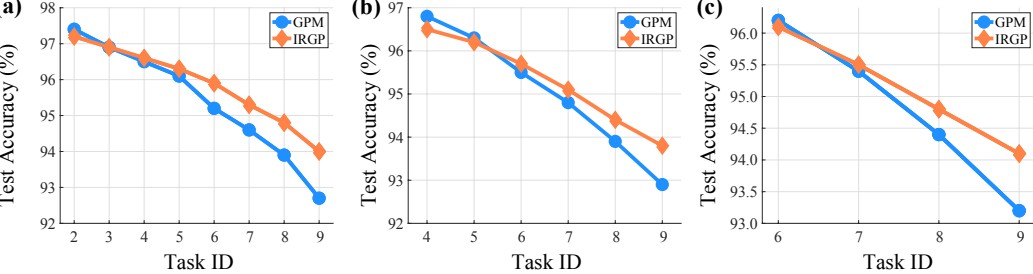

Figure 8: Accuracy evolution of the (a) 1st, (b) 3rd, and (c) 5th task on Mixture.

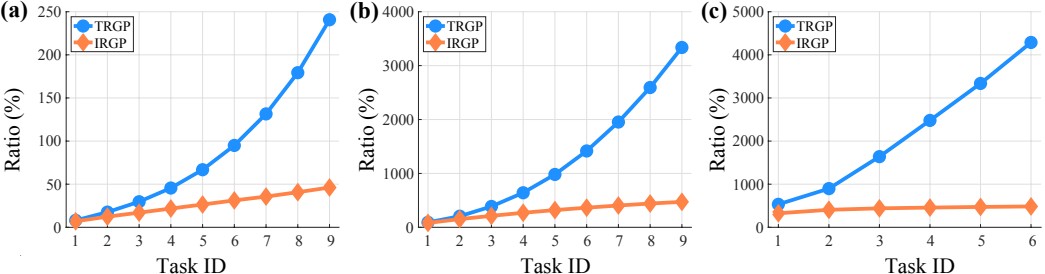

## C.4 TIME CONSUMPTION

We report the time consumption of IRGP on two benchmarks compared with relative baselines. TRGP and IRGP are both evaluated on a single NVIDIA GeForce RTX 2080 Ti GPU and we report the results according to (Lin et al., 2022). As discussed in Section 5, IRGP takes acceptable extra time compared with GPM on both datasets. For CIFAR-100 Split, IRGP tasks similar time as TRGP, which is similar to ER_Res and HAT and much less than A-GEM and OWM. For MiniImageNet, IRGP tasks around 30% time than TRGP, and is similar to A-GEM.

Table 14: Time comparison evaluated on two benchmarks. We use the results reported in (Lin et al., 2022) and the time is normalized with respect to GPM.

| Datasets | Methods | | | | | | | |
|---|---|---|---|---|---|---|---|---|
| | OWM | EWC | HAT | A-GEM | ER_Res | GPM | TRGP | IRGP |
| CIFAR-100 | 2.41 | 1.76 | 1.62 | 3.48 | 1.49 | 1.00 | 1.65 | 1.62 |
| MiniImageNet | - | 1.22 | 0.91 | 1.79 | 0.82 | 1.00 | 1.34 | 1.69 |

## C.5 MEMORY USAGE

We provide a comparison between TRGP and IRGP on the ratio of the amount of extra parameters concerning the amount of the parameters of the initial network architecture. According to Figure 9, TRGP requires at least 200% of the number of extra parameters after learning all tasks on the other three benchmarks, while IRGP only stores the representation of the frozen space, which can further be released in the inference phase.

Figure 9: Ratio of the amount of extra parameters concerning the amount of the parameters of the initial network architecture on (a) CIFAR100-Split, (b) PMNIST, and (c) Mixture.

## C.6 RELAXING RATIO

We provide the results on CIFAR100-Split between the hyper-parameter $\zeta$ and the relaxing ratio of all five layers of the AlexNet architecture in Table 15. And $\beta$ is set to be 1.0 for all experiments here. According to Table 15, generally, larger $\zeta$ guarantees smaller relaxing ratios. As discussed in Section 5, forgetting is mitigated by constraining the percentage of the relaxing weights, as a result of increasing $\zeta$.

Table 15: The relationship between $\zeta$ and the relaxing ratio of different layers on CIFAR-100 Split.

| $\zeta_{cong}$ | $\zeta_{fc}$ | Conv1 (%) | Conv2 (%) | Conv3 (%) | Fc1 (%) | Fc2 (%) | ACC | BWT |
|---|---|---|---|---|---|---|---|---|
| 0.20 | 0.20 | 72.32 | 61.56 | 69.11 | 38.92 | 29.75 | 71.97% | -3.0% |
| 0.50 | 0.50 | 50.96 | 45.18 | 56.07 | 27.43 | 19.06 | 72.58% | -2.3% |
| 0.80 | 0.80 | 50.99 | 31.09 | 35.74 | 10.56 | 5.41 | 73.34% | -2.2% |
| 0.90 | 0.90 | 44.91 | 23.05 | 21.90 | 4.82 | 2.00 | 73.28% | -1.2% |
| 0.95 | 0.90 | 42.07 | 17.82 | 16.87 | 4.84 | 2.20 | 73.52% | -0.9% |
| 0.95 | 0.95 | 42.77 | 17.49 | 16.59 | 2.30 | 0.76 | 73.08% | -0.6% |
| 0.99 | 0.99 | 33.68 | 7.08 | 8.07 | 0.16 | 0.00 | 73.19% | -0.6% |

## C.7 OTHER RESULTS

We provide the test accuracy over the task sequence in Figure 10. As shown in Figure 10-(a) and Figure 10-(b), our IRGP-Exp dominates TRGP on both benchmarks relaxing either 80% or T% of the frozen spaces under an expansion setting. We further compare IRGP with TRGP modified with the same regularization terms. As shown in Figure 10-(c) and Figure 10-(d), the performance of TRGP drops significantly constrained in a fixed network capacity on both benchmarks. Detailed results between IRGP and TRGP are provided in Table 16 and 17.

Figure 10: Test accuracy after learning each task under an expansion setting on (a) CIFAR-100 Split and (b) PMNIST, and within a fixed network capacity on (c) CIFAR-100 Split and (d) PMNIST.

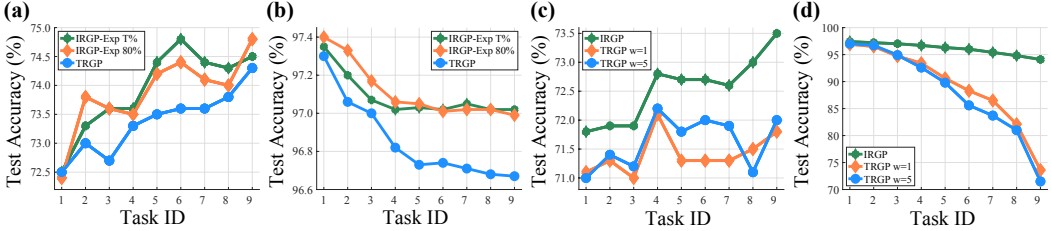

Table 16: Compare IRGP-Exp with TRGP on four benchmarks under an expansion setting. All results reported are average over 5 runs.

| Methods | TRGP | | IRGP-Exp 50% | | IRGP-Exp 80% | | IRGP-Exp T% | |
|---|---|---|---|---|---|---|---|---|
| | ACC (%) | BWT (%) | ACC (%) | BWT (%) | ACC (%) | BWT (%) | ACC (%) | BWT (%) |
| CIFAR | 74.46 | 0.07 | 75.15 | 0.20 | **75.38** | **0.29** | 75.06 | 0.04 |
| PMNIST | 96.34 | -0.80 | 96.68 | -0.61 | 96.99 | -0.38 | **97.03** | **-0.28** |
| MiniImageNet | 61.78 | -0.50 | 60.81 | -0.21 | **62.03** | 0.48 | 60.84 | **0.55** |
| Mixture | 83.54 | -0.80 | 82.45 | -0.74 | 83.22 | -0.48 | **83.62** | **-0.33** |

Table 17: Compare IRGP with TRGP-Reg on four benchmarks within a fixed network capacity. All results reported are average over 5 runs.

| Methods | IRGP | | TRGP-Reg $w = 1$ | | TRGP-Reg $w = 5$ | | TRGP-Reg $w = 50$ | |
|---|---|---|---|---|---|---|---|---|
| | ACC (%) | BWT (%) | ACC (%) | BWT (%) | ACC (%) | BWT (%) | ACC (%) | BWT (%) |
| CIFAR | **73.52** | **-0.94** | 71.85 | -1.48 | 72.08 | -1.51 | 72.46 | -0.96 |
| PMNIST | **94.20** | **-2.44** | 73.69 | -24.14 | 71.51 | -26.97 | 72.43 | -26.14 |
| MiniImageNet | **61.26** | -1.72 | 55.81 | -3.69 | 58.81 | -2.84 | 22.69 | **0.01** |
| Mixture | **77.91** | -4.45 | 73.31 | -11.05 | 74.71 | -9.33 | 17.36 | **-0.01** |

## C.8 FORWARD KNOWLEDGE TRANSFER: NEWLY-ADDED RESULTS FOR TABLE 2

Table 18: Comparison of forward knowledge transfer on CIFAR100-Sup, evaluated by $\Omega_{new}$.

| Methods | PGN | EWC | GPM | IRGP |
|---------|-----|-----|-----|------|
| $\Omega_{new}$ | $51.1 \pm 0.4$ | $57.5 \pm 3.1$ | $58.7 \pm 0.3$ | $\mathbf{58.9 \pm 0.4}$ |

# D   ALGORITHM

We present the pseudo-code of our *Iterative Relaxing Gradient Projection* here.

---

**Algorithm 2** Iterative Relaxing Gradient Projection

---

1:  Initiate frozen subspaces $\mathcal{U}_0 = \{U_0^l\}_{l=1}^L$ as $\emptyset$s and optimize $\mathcal{W}_1$ for task 1
2:  Compute frozen subspace $\mathcal{U}_1$ with Equation (4)
3:  **for** $t \in 2, ..., T$ **do**
4:      Initiate relaxing subspace $\{V_t^l\}_{l=1}^L$ as $\emptyset$s
5:      Initiate scaling matrices $\{\mathbf{S}_t^l\}_{l=1}^L$ as identity matrix $\mathbb{1}$s
6:      **repeat**
7:          Fine-tune for pre-defined $e_t$ epochs with objective function 11
8:          Search additional relaxing subspace $\{V_t^{l,new}\}_{l=1}^L$ by Algorithm 1 with current gradients
            $\{g_t^l\}_{l=1}^L$, remaining frozen subspace $\{U_t^{l,rest} = U_t^l \backslash V_t^l\}_{l=1}^L$ and thresholds $\{\epsilon_{th}^l, \gamma_t^l\}_{l=1}^L$
9:          $V_t^l \leftarrow V_t^l \cup V_t^{l,new}$
10:         Expand $\mathbf{S}_t^l$ with Equation (12)
11:     **until** $V_t^{l,new}$ is $\emptyset$
12:     Optimize $\mathcal{W}_t, \mathbf{S}_t$ with objective function 11
13:     Update $W_t^l \leftarrow W_t^l - \text{Proj}_{V_t^l}(W_t^l) + \text{Proj}_{V_t^l}^{S_t}(W_t^l)$
14:     Update frozen subspace $\mathcal{U}_t$ with Equation (4)
15: **end for**

---

