# OpenReview forum: "Iterative Relaxing Gradient Projection for Continual Learning"
_ICLR.cc/2023/Conference — Submitted to ICLR 2023_

### Official Review · Reviewer_tNh3 · 2022-10-23

**Confidence:** 4
**Correctness:** 2
**Technical Novelty And Significance:** 3
**Empirical Novelty And Significance:** 2
**Recommendation:** 6

**Clarity, Quality, Novelty And Reproducibility:**

The paper shows promising research directions. However, I still have a doubt about the definition of Forward Transfer that is being used.

I think the work has a good motivation: to take advantage of information that may be relevant to two tasks and consolidate that knowledge. Something that the authors present well, even though I have doubts as to how, how much and how many weights are moved/selected in this iterative process. It would be interesting to see how the authors' motivations to achieve a better transfer of knowledge between the different tasks are effectively fulfilled.

**Strength And Weaknesses:**

S:
- In order for CL to work, there must be motivation to add flexibility to the weights learned through previous tasks and that this flexibility causes minimal forgetting. It not only helps to consolidate learning, but it can also help with knowledge transfer. The motivation of the authors is interesting to the community.
- The idea of adding plasticity to the model by relaxing similar connections for both tasks (old and new) is interesting. Not only does it add plasticity to the current task, but it can also help with knowledge transfer to previous tasks.
    - Question To the authors: Do you study how the accuracy of previous tasks is affected when weights are relaxed?
- Table 2 provides helpful insights into how the flexibility of certain connections is used. Without making the model grow, favorable results are achieved.

W:
- The definition of Forward Transfer is not the same as the one used in the paper by Lopez-Paz and Ranzato. It is not clear to me if only the formula is incorrect or if the results are as well, but the way it is presented makes me think it is not only the formula. Forward Transfer has to do with the ability of the model to acquire knowledge that may be relevant to tasks that have not yet occurred. Here it is used for the current task.
    - Focusing on the accuracy of the current task is crucial, and not many papers focus on it. However, it is not the same as Forward Transfer.
- As the proposed method takes 60% more time than GPM, and only increases accuracy by 1%, it is not a favorable trade-off. It may depend on the application and scenario, but it is a big difference. However, I do appreciate that the authors mention this result.
- In Table 4, adding more extreme values would have been helpful. In order to determine whether the assumptions mentioned in the model are met, we must observe how the model behaves.

Questions:
- Since the community is focusing more (and for a good reason) on scenarios where the task-id is unavailable in inference. Do you think this method can be adapted to work in this type of scenario?
- Is it known what percentage of weights are relaxed in each task? Can this be related to forgetting the method?
- In the fifth line of the paragraph after Eq 1. Should U^l_{t-1} just be U_{t-1} or not?
- What is defined in Eq 9?
- In Eq 10, what is the I of W^l_{t,I}?
- Algorithm 2 is in Appendix D, not C.5


**Summary Of The Paper:**

The authors present a method to mitigate forgetting and favor forward transfer by using a gradient projection approach with a fixed-capacity architecture. The method (Iterative Relaxing Gradient Projection (IRGP)) iteratively searches for frozen (previously trained) weights that are relevant to the current task. These weights are relaxed to consolidate knowledge of the current task, which helps with one of the problems of gradient projection methods: low flexibility when training novel tasks. In addition, the authors present a version that can be expanded (increasing the number of weights in the model), which is efficient compared to a state-of-the-art method. Results superior to various methods are presented in different scenarios of Task-Incremental Learning, in addition to some ablation studies.

**Summary Of The Review:**

As the paper is, I don't think it's good to accept it. However, I think it is details that separate it from a paper that can be accepted.

The first is the definition of Forward Transfer and the second is a more complete study of how many and how the weights change when they are made flexible. If many weights are made flexible, one would expect more forgetfulness, if there are few, how do we ensure that there is indeed knowledge transfer?

---

> ### Author Response · Authors · 2022-11-11
> **Response**
>
> Thank you for the valuable comments and suggestions. We have uploaded a revision of our paper.
>
> **Question 1: The definition of FWT in Section 4.1 is incorrect.**
>
> Thanks for pointing this out. In this work, we focus forward knowledge transfer on the capability of acquiring new tasks by the test accuracy after learning each new task based on the observed tasks: $\Omega_{new} = \frac{1}{T-1} \Sigma_{i=2}^T (A_{i,i} - b_i)$ (Kemker et al., 2018). Some recent papers task $\Omega_{new}$ as the metrics for forward knowledge transfer, including CLEVA (Mundt et al., 2021) and CUBER (Lin et al., 2022), while FWT proposed by (Lopez-Paz & Ranzato et al. 2017) reflects the zero-shot performance on new tasks. We add the definition of $\Omega_{new}$ and amend the definition of FWT in Section 4.  To better demonstrate the performance of our method, we add results of FWT in Appendix C.2 as well. According to the results, our IRGP achieves better FWT on all three task-incremental benchmarks. Generally, our IRGP facilitates forward knowledge transfer by relaxing frozen weights.
>
> **Question 2: About the trade-off between time and accuracy.**
>
> As discussed in the paper, our IRGP is designed to facilitate forward knowledge transfer, especially $\Omega_{new}$, by relaxing the frozen spaces. For example, according to Table 8, IRGP gains 2.7% better $\Omega_{new}$ on CIFAR100-Split and around 1% better accuracy at the same time. Moreover, another strength of IRGP is its ability to work either under or without the constraint of a fixed network capacity, while GPM restricts the network capacity to be fixed, and TRGP requires additional capacity. This equips our IRGP with better adaptivity to different scenarios. In summary, while taking a moderately longer time to train, our IRGP achieves better forward knowledge transfer in addition to higher accuracy and is more adaptive in terms of network capacity.
>
> **Question 3: Better observation with extreme hyper-parameters.**
>
> Thanks for the suggestion, we add results of extremely large or small hyper-parameter $\zeta$ in Table 4 and Appendix C.6. According to Table 4 in the updated version of our paper, the accuracy and forgetting performance of IRGP degrades as decreasing $\zeta$ to 0.2, in accordance with our assumption.
>
> **Question 4: The relation between forgetting and the percentage of relaxing weights.**
>
> As the percentage of relaxing weights is constrained by hyper-parameter $\zeta$, we add the results of the relationship between $\zeta$, the percentage of relaxing weights and the forgetting in Table 14. According to Table 14, larger $\zeta$ guarantees less forgetting as a result of smaller percentage of relaxing weights, as discussed in Section 5.
>
> **Question 5: Adaptation to the class-incremental scenario.**
>
> Currently, IRGP mainly serves task-incremental and domain-incremental scenarios. As mentioned in GPM (Saha et al., 2021), methods of this category would perform better with small data replay under the class-incremental scenario. We leave the exploration of our method in the class-incremental scenario as future work.
>
> **Question 6: Whether $U^l_t$ in the fifth line of the paragraph after Eq. 1 is correct.**
>
> Our gradient projection strategy works in a layer-wise manner, constraining the gradients to be orthogonal to the frozen space of each layer. We have revised the line and aligned the notion of the gradients $g^l_t$ and the frozen space $U^l_t$ in a layer-wise manner.
>
> **Question 7: What is defined in Eq. 9?**
>
> In Eq. 9, we formulate the criterion of the relaxing space $V^l_t$. The upper equation guarantees all vectors within $V^l_t$ satisfy the angle threshold, while the lower equation guarantees there does not exist any vector within the rest space satisfying the angle threshold. This criterion is the same as $\underset{u \in V^l_t}{\max} \mathit{\Theta} ( u, R_{g,t}^l ) \leq \gamma^l_t$ and $\underset{v \in U_{t-1}^{l,c}}{\min} \mathit{\Theta} ( v, R_{g,t}^l ) > \gamma^l_t$ as mentioned below Eq. 9.
>
> **Question 8: What is the 'I' of $W^l_{t,I}$ in Eq. 10.**
>
> The 'I' of $W^l_{t,I}$ denotes 'Inference'. During the training phase, the weight $W^l_t$ works together with the scale matrices $S^l_t$. However, in the inference phase, the corresponding scale matrices $S^l_t$ are consolidated into the weight $W^l_t$. The consolidated weight parameters $W^l_t$ are denoted as $W^l_{t,I}$.
>
> **Question 9: Algorithm 2 is in Appendix D, not C.5.**
>
> We have corrected this in our revised paper.

---

> > ### Comment · Reviewer_tNh3 · 2022-11-16
> > **Response to authors Rebuttal**
> >
> > I appreciate the authors' detailed responses and the document's updating. The work is very interesting, the idea of improving the forward transfer is essential in CL models. However, for me, the forward transfer is an end to achieving other objectives: Better final accuracy, faster training, or better knowledge transfer. In this work, the authors partially achieve some of this final objective.
> >
> > Even though the proposal is incremental from previous works, it focuses on a clear limitation of those (Forward Knowledge Transfer) with a clever idea. Despite having results that do not clearly show the effect they want to propose and the limitations already presented, this work goes in the right direction.
> >
> > For these reasons, I increased my score.

---

> > > ### Author Response · Authors · 2022-11-16
> > > **Thank you**
> > >
> > > Thank you so much! We really appreciate your prompt response and insightful suggestions. We will further revise our paper with the added experiments and discussions.

---

### Official Review · Reviewer_Vp6m · 2022-10-24

**Confidence:** 3
**Correctness:** 3
**Technical Novelty And Significance:** 3
**Empirical Novelty And Significance:** 1
**Recommendation:** 6

**Clarity, Quality, Novelty And Reproducibility:**

As mentioned above, this paper heavily builds upon two recent papers (Saha et al., 2021, ICLR; Lin et al., 2022, ICLR). For example, the experimental benchmarks used and methods that are compared against are essentially the same. This has advantages in terms of reproducibility (as code for these benchmarks was provided by these previous papers), but only in a narrow sense, as it is unclear how well the proposed method would do in settings in which it and its predecessors were not developed.
This heavy reliance on these previous papers has also negative consequences in terms of novelty, and also in terms of clarity (the paper requires familiarity with these previous papers).

**Strength And Weaknesses:**

I think an important limitation of this paper is that it very closely follows the experimental setup of a couple of recent papers proposing similar methods, such as Saha et al. (2021, ICLR) and Lin et al. (2022, ICLR).

One resulting issue is that without knowledge of these two previous papers, it is very challenging, if not impossible, to appreciate the contribution of the current paper. I would encourage the authors to rewrite their paper to make it possible to appreciate its contribution without knowledge of these previous papers.

Another issue resulting from this is that there is an important risk of overfitting on the specific benchmarks used. On the one hand (e.g., for reproducibility) it is great that the authors include experiments on these extensively studied benchmarks, but as for several iterations now this line of work has exclusively been evaluated on this set of benchmark, I believe there is a real risk of overfitting and I would encourage the authors to evaluate their method also on other benchmarks.

Another consequence of sticking so closely to these previous papers, is that the set of methods that is compared against is exactly the same as in those previous papers. Other recent related methods are not considered. In particular, one very related method that is not considered is Natural Continual Learning proposed by Kao et al. (2021, Neurips; https://arxiv.org/abs/2106.08085). This method also tackles continual learning by putting restrictive constraints on the optimization of new tasks, but, unlike the present paper, uses an approximation of the Fisher Matrix of old tasks for this. Intuitively, this should allow for more positive transfer than using an orthogonal projection onto the input subspaces of olds tasks. I strongly encourage the authors to compare against this paper.

In the continual learning literature, often a distinction is made between task-incremental learning, domain-incremental learning and class incremental learning (e.g., as in Kao et al., 2021, NeurIPS; or see https://arxiv.org/abs/1904.07734). The current paper does not discuss these distinctions, and it is not entirely clear to which of these three scenarios of continual learning the considered benchmarks belong. My guess would be that PMNIST follows the assumptions of domain-incremental learning, while the others follow task-incremental learning? It would help the clarity of the paper to make this clear.

Other issues:
- In section 2, why is GEM not discussed under gradient projection methods? It seems clear that this method is doing gradient projection as well.
- Whenever experimental results are copied over from another paper, this should be clearly explained in both the text and the legend of the table/figure where those results are reported.
- I think it is important to include the standard deviations in the tables in the main text. Currently they are only provided in the Appendix, but for a correct interpretation of the results they are essential.
- In Table 1, it is misleading that the BWT results of “Ours (IRGP)” are printed in bold. For none of the considered benchmarks this method scores the best on this metric.
- For AGEM and ER_Res it should be stated in the main text what memory buffer size is used.


**Summary Of The Paper:**

To tackle the problem of incrementally learning a sequence of tasks with a deep neural network, this paper builds upon and extends a recent line of work using gradient projection to constrain the optimization of newly encountered tasks. In particular, this paper builds upon the TRGP method proposed by Lin et al. (2022, ICLR). A disadvantage of TRGP is that, when learning new tasks, TRGP expands the model with a new trust region (and associated parameter storage) for each new task. To avoid such expansion, the IRGP method proposed in this paper instead iteratively searches for the most critical parameter subspace to maintain. This results in a method that still performs well on a specific set of benchmarks (approaching the performance of TRGP, and if also using additional parameters it can outperform it) with less rapidly increasing storage costs.

**Summary Of The Review:**

Although I believe this paper probably makes a minor contribution to the literature, the restricted experimental evaluation (in particular the heavy reliance on previous papers) causes me to currently recommend rejection. I encourage the authors to test their idea in more varied settings, compare their idea to other related methods and to improve the writing of the paper to reduce the reliance on knowledge of other papers.

---

> ### Author Response · Authors · 2022-11-11
> **Response**
>
> Thank you for the helpful comments and suggestions. We have uploaded a revision of our paper.
>
> **Question 1: Why follow the experimental setup in GPM and TRGP? This leads to a sophisticated presentation.**
>
> Our method is built upon the gradient projection methods GPM and TRGP, which show good performance on accuracy by constraining the gradient directions. However, their forward knowledge transfer ability is hindered at the same time, which can be attributed to the frozen weights of the previous tasks. Having observed this, we proposed IRGP to relax the frozen weights iteratively, which facilitates forward knowledge transfer with flexible constraints while achieving better accuracy at the same time. Besides, we have also sketched the main difference between our method and the two baselines in Figure 1. We will continue to refine the presentation to make our work easier to understand.
>
> **Question 2: Considering Natural Continual Learning as a baseline.**
>
> Thanks for introducing this work. We've added the summarized idea of NCL in the related work section: NCL introduces Baysian weight regularization to optimize for the trust region computation to mitigate catastrophic forgetting. In contrast, IRGP focuses on directly relaxing the frozen trust region to facilitate forward knowledge transfer. These regularization terms are introduced to increase flexibility in IRGP, instead of better trust region optimization. Moreover, on benchmarks such as CIFAR100-Split, NCL works with pre-trained model parameters, which currently are not publicly released. Therefore, while NCL is undoubtedly an inspiring work, we leave the exploration of comparison and combination of IRGP and NCL as crucial future work.
>
> **Question 3: Overfitting on specific benchmarks from (Saha et al., 2021)**
>
> We follow (Saha et al., 2021) to use the three traditional benchmarks, CIFAR100-Split, PMNIST, and MiniImageNet, to facilitate a fair comparison with the baselines. However, the Mixture dataset we used, which originated from (Serra et al., 2018), was evaluated by neither GPM nor TRGP in their papers. As mentioned in Section 4 and Appendix B.1, we reimplemented the algorithm and evaluated all baselines on this benchmark. According to the results reported in Table 1, IRGP achieves better accuracy in addition to forward knowledge transfer over all benchmarks.
>
> **Question 4: Clarify the three learning scenarios.**
>
> Thanks for the suggestion and we add the interpretation in the updated version of the paper. PMNIST works under a domain-incremental scenario while others work under a task-incremental scenario.
>
> **Question 5: Categorizing GEM as gradient projection methods in Section 2.**
>
> Although GEM constrains the direction of the gradients, it requires an extra memory buffer. Thus, we introduce GEM in the *replay-based methods* section. In the updated version of our paper, we clarified this in the *gradient projection methods* section as well.
>
> **Question 6: The results of the standard deviation should be reported in the main text.**
>
> Thanks for your suggestion. In order to demonstrate our results better, we provided the standard deviation results in Table 1 instead of in the Appendix.
>
> **Question 7: The sources of the experimental results should be clearly explained.**
>
> As mentioned in the first paraphrase of Section 4.2, the experiments on the Mixture dataset are implemented by us and other results are from (Saha et al., 2021). Further, to clarify the sources better, we have also reorganized the expression in the updated version of our paper.
>
> **Question 8: The highlighted results in Table 1 are misleading.**
>
> We have corrected the highlighting in Table 1 by printing the best scores in bold and the second-best scores with underlines.
>
> **Question 9: The buffer size of A-GEM and ER_Res shoulb be stated in the main text.**
>
> We revised our paper and highlighted the buffer size in Section 4.1 and Appendix B.3. Following (Saha et al., 2021), we use 1000, 2000, 500, and 3000 samples as the memory buffer size for A-GEM and ER_Res on PMNIST, CIFAR-100 Split, MiniImageNet, and Mixture respectively.

---

> > ### Comment · Reviewer_Vp6m · 2022-11-15
> > **Response to author rebuttal**
> >
> > Thanks to the authors for the rebuttal and the changes to the paper. These changes have clarified certain aspects and improved the paper. I have increased my score as a result. I think this paper makes a contribution, but, as currently presented and evaluated, this contribution is still somewhat incremental on top of the works Saha et al. (2021, ICLR) and Lin et al. (2022, ICLR).
> > I would still like to encourage the authors to test their idea in more varied settings and to compare their idea to other related methods. The authors use as argument not to compare against NCL that this method "works with pre-trained model parameters", but I can not see an issue with running that method without pre-trained model parameters.

---

> > > ### Author Response · Authors · 2022-11-16
> > > **Thank you**
> > >
> > > Thank you for updating the score! Your feedback really helped us improve our work. We will continue to improve our work by comparing it with related methods in more varied settings.

---

### Official Review · Reviewer_AinZ · 2022-10-30

**Confidence:** 4
**Correctness:** 2
**Technical Novelty And Significance:** 2
**Empirical Novelty And Significance:** 2
**Recommendation:** 5

**Clarity, Quality, Novelty And Reproducibility:**

Clarity. The paper is well-written and easy to follow,.
Novelty. The work is novel.
Reproducibility. All details should be provided in this paper. The paper follows the experimental setup of previous work providing a citation.

**Strength And Weaknesses:**

Strengths
+ Interesting solution to the stability-plasticity dilemma for gradient projection continual learning methods.
+ Clear mathematical exposition of the related and proposed methods.

Weaknesses
- Although IRGP was introduced to facilitate forward knowledge transfer, the metric FWT is only reported in the appendix and compared with only one method. This is clearly unacceptable. FWT should be reported in Tables 1-3.
- The proposed method is an iterative method, yet there is no mention or analysis of computational complexity or resources are mentioned anywhere in the paper let alone compared with existing work.
- The experiments are not consistent. It is difficult to get a clear view on how the method really compares especially versus expansion-based methods because only one benchmark is reported. Why only CIFAR-100 or PMNIST?

**Summary Of The Paper:**

This paper proposes Iterative Relaxing Gradient Projection (IRGP) to facilitate forward knowledge transfer within a fixed network capacity for continual learning.

**Summary Of The Review:**

The proposed method is interesting but the empirical evaluation is not convincing.

Update. I thank the authors for the response and addressing many of my and other reviewers' comments. I have updated the score to reflect the recent changes. The reason for not providing a higher score is current missing results due to the time constraint.

---

> ### Author Response · Authors · 2022-11-11
> **Response**
>
> Thank you for the constructive suggestions and comments. We have uploaded a revision of our paper.
>
> **Question 1: Lacking results of forward knowledge transfer in Table 1-3.**
>
> Previously, we validate the improved performance of forward knowledge transfer in Figure 2-\(c\) and Appendix C.2. In the revision, we follow the suggestion to report the forward transfer results in the main tables. For Table 2, and the {CIFAR-100 Split, MiniImageNet, and PMNIST benchmarks} in Table 1, however, we were using the accuracy results reported in the GPM paper (Saha et al., 2021) for direct comparison. We are now rerunning these experiments to evaluate their forward knowledge transfer performance. Due to the time limit, for now, we report the forward transfer performance on the Mixture benchmark over all baselines in Table 12 (also see the following table). It can be seen that our method achieves better forward knowledge transfer than all the other baselines that do not require an extra memory buffer, which is consistent with the analysis in Section 4. We will complete the results and report them in the main paper when we finish re-running all experiments.
>
> | Method        | A-GEM          | ER_Res             | EWC            | HAT            | GPM            | IRGP               |
> |:---------------:|:----------------:|:--------------------:|:----------------:|:----------------:|:----------------:|:--------------------:|
> | Memory Buffer | Yes            | Yes                | No             | No             | No             | No                 |
> | Mixture       | 85.9 $\pm$ 0.4 | **86.4 $\pm$ 0.2** | 74.4 $\pm$ 1.0 | 79.3 $\pm$ 0.2 | 82.1 $\pm$ 0.4 | **82.2 $\pm$ 0.4** |
>
> For Table 3, we have additionally reported forward knowledge transfer performance in all experimental settings on all benchmarks (see the revised Table 3). It is shown that IRGP achieve improved forward transfer and accuracy performance than TRGP in most experimental settings and benchmarks, whether under or without the constraint of a fixed network capacity.
>
> **Question 2: Lacking comparsion of computational complexity and resources.**
>
> In fact, we have reported the training time comparison among all methods on CIFAR100-Split and MiniImageNet in Appendix C.4. According to Appendix C.4, IRGP takes comparable or smaller time compared with regularization-based EWC and runs consistently faster than replay-based A-GEM. In addition, our method requires acceptable extra time compared with GPM.
>
> **Question 3: Lacking experiments on other benchmarks versus expansion-based methods.**
>
> As recommended, we add the results on the MiniImageNet and Mixture benchmarks in Table 3 of our revised paper. Detailed results are also provided in Tables 16 and 17 in Appendix. The results demonstrate that our method consistently outperforms TRGP with/without expanding the network architecture, which is in accordance with the results in CIFAR100-Split and PMNIST.

---

> ### Author Response · Authors · 2022-11-19
> **Thank you and newly-added results**
>
> Thanks so much for your insightful suggestions and for updating the score!
> We have just completed evaluating the forward transfer performance of the other three benchmarks in Table 1. We added these results in Table 12 of the newly-updated paper. According to Table 12, our IRGP consistently achieves the best or the second-best performance without extra replay buffers.
>
> | Method | A-GEM | ER_Res | EWC | HAT | GPM | IRGP |
> |:---------------:|:----------------:|:--------------------:|:----------------:|:----------------:|:----------------:|:--------------------:|
> | Memory Buffer | Yes            | Yes                | No             | No             | No             | No                 |
> | PMNIST       | **97.4 $\pm$ 0.0** | 97.4 $\pm$ 0.5 | 93.1 $\pm$ 0.7 | - | **96.6 $\pm$ 0.0** | \[*96.5 $\pm$ 0.1*\] |
> | CIFAR100-Split       | **77.5 $\pm$ 0.4** | 77.1 $\pm$ 0.2 | 69.9 $\pm$ 1.1 | \[*71.5 $\pm$ 0.6*\] | 71.5 $\pm$ 0.5 | **74.1 $\pm$ 0.2** |
> | MiniImageNet       | 67.6 $\pm$ 1.2 | **69.5 $\pm$ 0.4** | **63.5 $\pm$ 2.8** | 62.6 $\pm$ 0.6 | 61.6 $\pm$ 0.6 | \[*62.8 $\pm$ 0.8*\] |
> | Mixture       | 85.9 $\pm$ 0.4 | **86.4 $\pm$ 0.2** | 74.4 $\pm$ 1.0 | 79.3 $\pm$ 0.2 | \[*82.1 $\pm$ 0.4*\] | **82.2 $\pm$ 0.4** |
>
> Moreover, for Table 2, we report the forward knowledge transfer performance of IRGP with comparisons to that of GPM, EWC, and PNN on CIFAR100-Sup (Table 18; also see the following table). IRGP gains better forward knowledge transfer than both GPM and EWC. Currently, we are working on other expansion-based baselines. We will complete the results in Table 13 after we finish re-running all experiments.
>
> | Method        | PNN            | EWC            | GPM            | IRGP               |
> |:---------------:|:----------------:|:----------------:|:--------------------:|:--------------------:|
> | CIFAR100-Sup       | 51.1 $\pm$ 0.4 | 57.5 $\pm$ 3.1 | 58.7 $\pm$ 0.3 | **58.9 $\pm$ 0.4** |

---

### Decision · Program_Chairs · 2023-01-20

**Decision:**

Reject

**Justification For Why Not Higher Score:**

Even after the rebuttal, all reviewers are still concerned around the combination of limited novelty and empirical results. All reviewers agreed that this paper needs more work.

**Justification For Why Not Lower Score:**

N/A

**Metareview: Summary, Strengths And Weaknesses:**

This paper proposes a method for achieving continual learning in the presence of multiple tasks by reducing knowledge forgetting while encouraging forward transfer. The key idea is to avoid augmenting the model with new parameters by iteratively searching and relaxing the most effective trust region.

The reviewers found the paper to be well written and also consider the main idea (especially positive transfer) to be interesting and valid. It is also notable that the method ensures that forgetting is minimized without the model having to grow.

I would like to note that although this paper received only three reviews, there has been reasonable discussion between authors and reviewers and a lot of discussion privately among reviewers and the AC (even outside of openreview). These discussions resulted in the authors to increase their score. However, it seems that after discussing with the reviewers the consensus is that the weaknesses outweigh the strengths of the paper.

In particular, the proposed method follows very closely [Saha et al., 2021, ICLR; Lin et al., 2022, ICLR], meaning that the work is rather incremental. At the same time, the reviewers believe that the experimental results are insufficient and at some cases weak (e.g. results for positive transfer). For example, further benchmarks or baselines should be considered. It is understood that this paper builds and therefore compares against two particular pieces of prior work, however, it would add more value to the community if the method's merits were demonstrated more independently. After all, the reader will be more concerned about how this method can be used to solve the continual learning problem in general, independently of which other methods served as inspiration to this one.


**Summary Of Ac-Reviewer Meeting:**

During our discussion with the reviewers there was noone willing to champion it. One reviewer with score 6 mentioned that their score is really between 5 and 6. Both reviewers who gave a score of 6 are still concerned around the combination of limited novelty and empirical results. In essence, the scores that were slightly above the borderline were in both cases motivated by the demonstration of positive transfer, however it is clear to me that we have reached a consensus that this is not enough to outweigh the weaknesses of the paper, and we hope that an improved version of this manuscript will be submitted in the near future.